# The Oligodendrocyte Transcription Factor 2 OLIG2 regulates transcriptional repression during myelinogenesis in rodents

Kunkun Zhang[1,2,3,5], Shaoxuan Chen[1,2,3,5], Qihua Yang[3], Shuanghui Guo[3], Qiang Chen[1,2,3], Zhixiong Liu [1,2,3], Li Li[3], Mengyun Jiang[3], Hongda Li[3], Jin Hu[3], Xu Pan[4], Wenbo Deng [2], Naian Xiao[1,2], Bo Wang[3], Zhan-xiang Wang[1,2 ✉], Liang Zhang [1,2,3 ✉] & Wei Mo [1,2,3 ✉]

OLIG2 is a transcription factor that activates the expression of myelin-associated genes in the oligodendrocyte-lineage cells. However, the mechanisms of myelin gene inactivation are unclear. Here, we uncover a non-canonical function of OLIG2 in transcriptional repression to modulate myelinogenesis by functionally interacting with tri-methyltransferase SETDB1. Immunoprecipitation and chromatin-immunoprecipitation assays show that OLIG2 recruits SETDB1 for H3K9me3 modification on the *Sox11* gene, which leads to the inhibition of *Sox11* expression during the differentiation of oligodendrocytes progenitor cells (OPCs) into immature oligodendrocytes (iOLs). Tissue-specific depletion of *Setdb1* in mice results in the hypomyelination during development and remyelination defects in the injured rodents. Knockdown of *Sox11* by siRNA in rat primary OPCs or depletion of *Sox11* in the oligodendrocyte lineage in mice could rescue the hypomyelination phenotype caused by the loss of OLIG2. In summary, our work demonstrates that the OLIG2-SETDB1 complex can mediate transcriptional repression in OPCs, affecting myelination.

[1] Department of Neurosurgery and Department of Neuroscience, the First Affiliated Hospital of Xiamen University, State Key Laboratory of Cellular Stress Biology, School of Medicine, Xiamen University, Fujian, China. [2] Xiamen Key Laboratory of Brain Center, the First Affiliated Hospital of Xiamen University, Fujian, China. [3] School of Life Sciences, Innovation Center for Cell Signaling Network, Xiamen University, Fujian, China. [4] School of Pharmaceutical Sciences, Xiamen University, Xiamen, Fujian 361102, China. [5]These authors contributed equally: Kunkun Zhang and Shaoxuan Chen. ✉email: wangzx@xmu.edu.cn; lzhangxmu@xmu.edu.cn; wmo@xmu.edu.cn

The master regulators refer to transcriptional factors determining the cell fate by dictating transcriptome, which has been well documented during reprogramming process. Extensive activation and inactivation of gene expression also occur in programmed differentiation[1]. Oligodendrocytes accelerate the conduction of nerve impulse by myelinating neuronal axons in the central nervous system (CNS)[2,3]. Dysregulated myelination is associated with various devastating neurological diseases, such as multiple sclerosis and leukodystrophy[4–6]. OPCs differentiate into iOLs that subsequently develop into mature oligodendrocytes (mOLs). Previous work has well painted the gene activation networks that promotes OPCs differentiation[7,8]. However, the importance and mechanism of gene down-regulation in each stage for myelinogenes is were not fully understood.

OLIG2 is a master regulator for oligodendrocyte lineage specification, especially regulating key stages of early oligodendrocyte development[9–13]. OLIG2 has been reported to be a upstream regulator of SOX10, another master regulator in oligodendrocyte development[13]. In addition, it has been demonstrated that OLIG2 recruits SWI/SNF chromatin remodeling enzyme SMARCA4 (also known as BRG1) to potentiate the expression of several myelin-related factors to initiate OPCs differentiation into iOLs[7]. Interestingly, OLIG2 has been reported to determine the cell fate of motor neurons, but acts as a gene repressor[14–16]. During the switching process of cell fate, master regulators genome-widely reconstruct epigenetic patterns, which subsequently guides the new transcriptomes[17]. H3K9me3 modification is one of the most predominant approaches for gene deactivation epigenetically, which is vital in embryonic development and reprogramming[18]. Deficiency of SETDB1, a major tri-methyltransferase of H3K9me3 leads to stem cell necroptosis in gut[19].

Here, we show a subset of genes that should be repressed to guarantee the OPC differentiation and myelination program are suppressed by a transcriptional inhibition machinery consisting of OLIG2 and SETDB1. OLIG2 guides and reloads the H3K9me3 deposition along the genome to promote the generation of iOLs from OPCs. Disrupting OLIG2-SETDB1 complex in the early OL lineage causes hypomyelination in mice. Our findings disclose a distinct mechanism that OLIG2 composed a TF-Epigenetic complex to conduct gene inhibition, for OL cell fate determination and myelination.

## Results

### The repressive role of OLIG2 is essential for oligodendrocyte differentiation

The differentiation process of OPCs can be divided into early (OPC- iOL) and late (iOL- mOL) stages. OLIG2 recruits BRG1 for chromatin remodeling, which activates gene network required for OPC to iOL transition[7]. Analyses of two published databases (transcriptomes[20] and OLIG2 bind sites[7]) revealed that, unexpectedly, enhanced olig2 binding also led to also led to repressed gene expression and the number of downregulated genes were comparable with that of upregulated genes (Fig. 1a), which implicates a non-canonical function of OLIG2 in transcriptional inhibition in oligodendrocytes. 50% of the downregulated genes in the early stage were intensively bound with OLIG2, while the ratio decreased to 30% in the late stage (Supplementary Fig. 1a), suggesting that OLIG2 is more dominant for gene repression in early stage. Gene Ontology (GO) analyses showed that downregulated genes with increased OLIG2 binding in early stage are enriched for those that function in cell growth and cell migration (Fig. 1b).

To determine whether the repressive role of OLIG2 is essential for myelination, a dominant-active form of OLIG2 (OLIG2 fused to 4X transactivation domain of VP16, referred to as OLIG2-VP64) (Supplementary Fig. 1b) was used to override endogenous OLIG2 in rat OPC in vitro. OLIG2-VP64, the dominant-active form of OLIG2, is reported in prior studies to abolish the transcriptional inhibition by transforming OLIG2 into transcriptional activation form[15,21]. Consistently, most of the differentially expressed genes (DEGs) were upregulated in cells transfected with OLIG2-VP64 (Supplementary Fig. 1c). Upon triiodothyronine (T3) stimulation for promoting differentiation, OPCs transfected with OLIG2-VP64 failed to exhibit morphological features of mOLs (Fig. 1c). Expression of Myelin marker proteins such as CNP or MBP was found in fewer cells expressing OLIG2-VP64 (Fig. 1d) and the expression levels of these marker genes were also decreased in these cells (Fig. 1e). Note that overexpressing VP64 alone had no appreciable effect on the OPC differentiation (Supplementary Fig. 1d). Transcriptomic analyses revealed that genes programmed to be inhibited by OLIG2 upon differentiation were upregulated in the OLIG2-VP64 overexpressed cells (Fig. 1f). To assess the role of OLIG2 in OPC differentiation in vivo, we injected with retrovirus containing either Cre-T2A-Olig2-T2A-GFP or Cre-T2A-Olig2-VP64-T2A-GFP into the corpus callosum of Olig2$^{Flox/Flox}$ mice at postnatal day 5 (P5) (Supplementary Fig. 1e). Endogenous Olig2 was deleted by Cre recombinase, which was reconstituted with exogenous OLIG2 or OLIG2-VP64. We collected the GFP$^+$ cells (infected cells) from animals at P15 for cell fate analyses. (Fig. 1g). 60% of OPCs reconstituted with OLIG2 successfully differentiate into CC1$^+$ oligodendrocytes. In contrast, less than 10% of OPCs reconstituted with OLIG2-VP64 were CC1$^+$ (Fig. 1h, i). Taken together, the repressive role of OLIG2 is required for myelinogenesis.

### OLIG2 assembles a repressor complex to dominate oligodendrocyte differentiation

Considering that OLIG2 has dual roles in both gene activation and deactivation, it is feasible to break down OLIG2-mediated inhibitory complex rather than wipe OLIG2 itself off. Master regulators often cooperate with epigenetic factors to regulate gene expression. We took a proteomics approach to identify the key components that binds to Olig2 for transcriptional regulation (Supplementary Fig. 2a). The capture of known coactivator BRG1 highlights the reliability of proteomic experiments (Fig. 2a). DNA or histone methylation triggers gene deactivation. Enzymes fulfilling DNA methylation such as DNMT1 or DNMT3a were not found to be interacting with OLIG2; but factors of histone modification were enriched (Fig. 2b). H3K9 or H3K27 methylation results in gene repression. Enzymes and cofactors mediating H3K9 but not H3K27 methylation were identified, such as SETDB1, TRIM28 (also known as KAP1), EHMT1 (also known as GLP), EHMT2 (also known as G9a), etc (Fig. 2a)[22,23]. That H3K9 but not H3K27 methylation is mentioned for regulation of OPC differentiation by Liu et al.[24]. If OLIG2 inhibits gene expression through H3K9 methylation to promote myelination, then disrupting H3K9 methylation will also affect the myelination. Knockdown experiments (Supplementary Fig. 2b, c) was performed in the primary rat OPCs to identify the most important enzymes for myelination. Silencing SETDB1, but not other epigenetic factors, not only led to markedly decreased expression of Mag and Mbp, makers of myelin (Fig. 2c), but also significantly impeded the efficacy of oligodendrocyte generation (Fig. 2d).

SETDB1 is a tri-methyltransferase of H3K9. The binding between OLIG2 and SETDB1 was validated by immunoprecipitation (Supplementary Fig. 2d, e). As Ehmt2 is a specific H3K9me2 (but not H3K9me3) transferase according to previous studies[25], we further examined if interactions between OLIG2 and EHMT1, SUV39h1 or PRDM2 exist, and we hardly detected any (Supplementary Fig. 2f). We further checked the integrity of

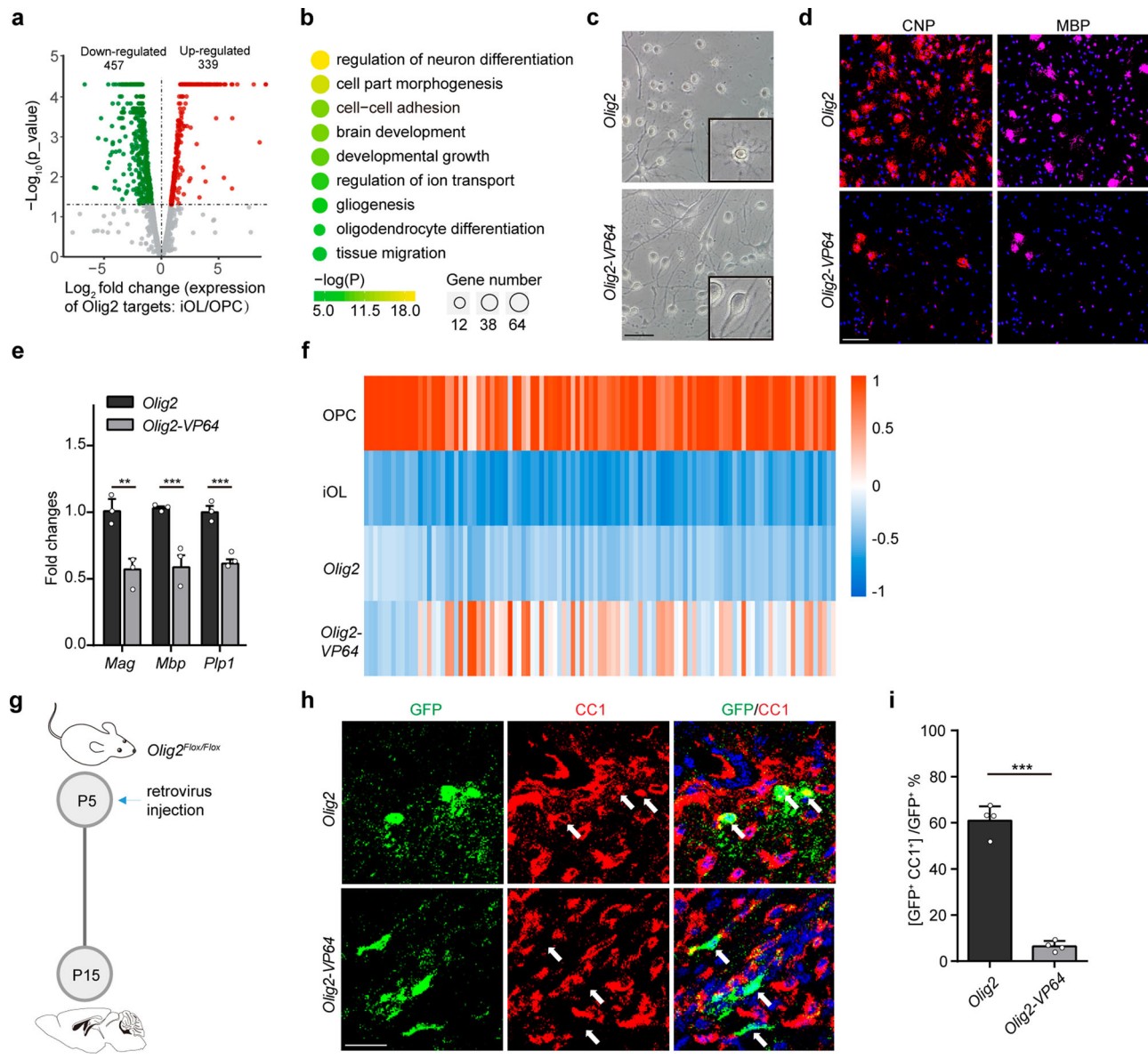

**Fig. 1 The repressive role of OLIG2 is essential for oligodendrocyte differentiation. a** Volcano plot of differential expressed genes (DEGs, log$_2$FC ≥ 1 or ≤ -1; $p$ value <0.05) that are targeted by enhanced OLIG2 binding in rat OPCs to iOL transition. **b** Representative GO analysis of the significantly downregulated genes in Fig. 1a. **c**, **d** Representative phase micrographs(c) and CNP$^+$ and MBP$^+$ immunostaining (d) in rat OPCs transfected with *Olig2* or *Olig2-VP64* under differentiation condition. $n$ = 3 independent experiments. **e** Quantitative real-time PCR analysis of oligodendrocyte differentiation-associated genes in rat OPCs transfected with *Olig2* or *Olig2-VP64* under differentiation condition. $n$ = 3 independent experiments. Error bars indicate SEM (*Mag*, $p$ = 0.0084; *Mbp*, $p$ < 0.001; *Plp*, $p$ < 0.001. two-tailed unpaired Student's $t$ test). **f** Heat map representing the expression of OL differentiation–related genes in rat OPC, iOL, iOL transfected with OLIG2 and iOL transfected with OLIG2-VP64. **g** Schematic diagram for retrovirus injection. **h** Immunostaining for CC1 on OLIG2 or OLIG2-VP64 retrovirus infected brain at P15. Arrow indicates the cell infected with retrovirus. $n$ = 4 control and 4 *Olig2-VP64* mice. **i** Quantification of the percentage of differentiated oligodendrocytes in infected cells. Error bars indicate SEM ($p$ < 0.001. two-tailed unpaired Student's $t$ test). Scale bars: 50 μm in (**c**); 100 μm in (**d**); 25 μm in (**h**).

OLIG2-BRG1 transcriptional activation complex in Setdb1 knockdown rat iOLs. And we observed that the interaction between OLIG2 and BRG1 was barely affected in the absence of SETDB1 (Supplementary Fig. 2g). In mouse brain, protein levels of SETDB1 and H3K9me3 were higher in differentiated oligodendrocytes (GSTπ$^+$ or CC1$^+$) than that in OPCs (OPC: PDGFRα$^+$)(Supplementary Fig. 2h, i). Notably, SETDB1 expression decreased rapidly in fully matured oligodendrocytes (iOL: CC1$^+$/MBP$^-$; mOL: CC1$^+$/MBP$^+$) (Fig. 2e), which suggests that SETDB1-mediated H3K9me3 modification mainly functions in the early stage. Consistently, the interaction of OLIG2 with SETDB1 was most prominent in P7 mouse forebrain when

massive OPCs are differentiating (Fig. 2f), which was consistent with the temporary formation of OLIG2-SETDB1 complex during OPC-iOL period when increased enrichment of H3K9me3 modification was observed in the genome (Fig. 2g). We further explored the fluctuation of SETDB1 levels in OPC lineage cells from young and old mice. As the result shown, SETDB1 decreased dramatically as mice age (Supplementary Fig. 2j). Cutting off the common H3K9me3 binding sites, the distribution of H3K9me3 enriched regions in gene body were more abundant in rat iO than that in OPCs (Fig. 2h). Genes that surrounded with iOL-specific H3K9me3 enrichment regions were very similar with genes that are downregulated upon OPCs

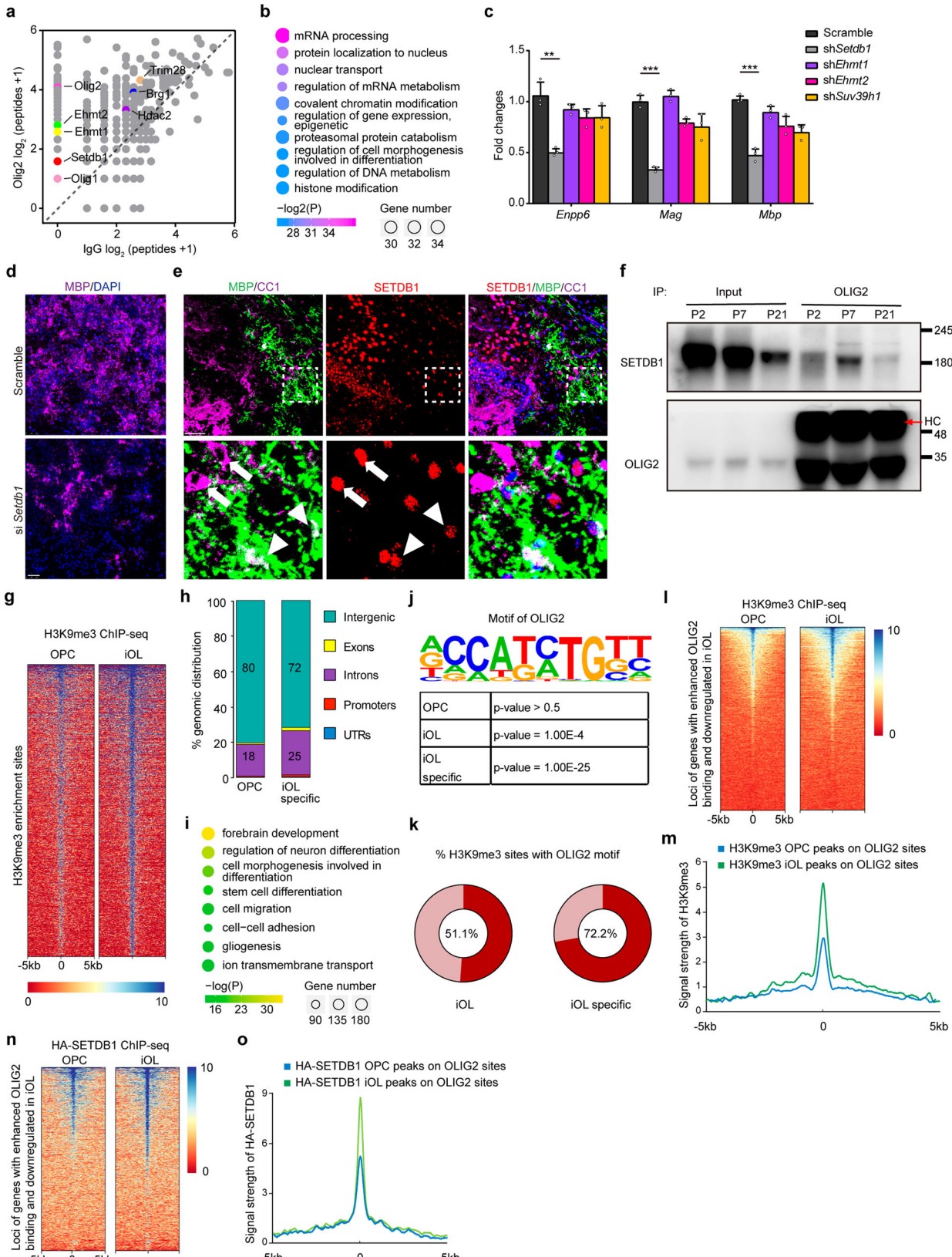

differentiation with increased OLIG2 binding (Figs. 2i and 1b), indicating the downregulation of those genes are likely coordinated by OLIG2 and H3K9me3. Motif analysis of the H3K9me3 enrichment regions was performed, which revealed the consensus binding motif for OLIG2 in iOLs but not in OPCs (Fig. 2j). 72.2% iOL-specific H3K9me3-enrichment regions, which do not show up in OPCs, contained OLIG2 motif, while the ratio reduced to 51.1% when we considered whole H3K9me3-enrichment regions in iOLs (Fig. 2k). Furthermore, in rat iOLs the enrichment of H3K9me3 was strengthened in loci of genes which are downregulated and have enhanced OLIG2 binding upon differentiation (Fig. 2l, m). The occupancy of SETDB1 along genome was

**Fig. 2 OLIG2 assembles a repressor complex to dominant oligodendrocyte differentiation. a** Scatter plot of proteins bound to OLIG2. Enriched epigenetic associated protein especially H3K9m3 modifiers were highlighted. **b** Representative GO analysis of protein pulled down with OLIG2. **c** Expression of *Enpp6*, *Mag* and *Mbp* in rat OPCs transfected with indicated shRNA under differentiation condition. Error bars indicate SEM (*Enpp6*, $p = 0.0031$; *Mag*, $p < 0.001$; *Mbp*, $p < 0.001$. two-tailed unpaired Student's *t* test). **d** Immunolabeling of MBP in rat OPCs transfected with indicated siRNA followed by differentiation-induction. $n = 3$ independent experiments. **e** Brain at P7 was immunostained with MBP, CC1 and SETDB1. Boxed areas in upper panels are shown at a high magnification in lower panels. Arrow indicates the co-expression of SETDB1 with CC1 but not MBP, and arrowhead indicates the co-expression of SETDB1 with CC1 and MBP. $n = 3$ different mice. **f** Co-immunoprecipitation (co-IP) of endogenous OLIG2 with SETDB1 in mouse cortex at P2, P7 and P21, respectively. HC heavy chain. $n = 3$ independent experiments. **g** Heatmap of H3K9me3- enrichment signals in rat OPCs and iOLs. **h** Genomic distribution of H3K9me3 enrichment regions in rat OPCs and specific in iOLs. **i** Representative GO analysis of genes nearest to iOL-specific H3K9me3 enrichment regions. **j** Binding motif identified by HOMER with H3K9me3- enrichment peaks. Letter size indicates nucleotide frequency at each position of OLIG2 binding motif. **k** The percentage of H3K9me3- enrichment peaks with OLIG2 binding motif in rat iOL. Heavy red color means the ratio. **l, m** Heatmap (**l**) and enrichment profiles (**m**) of H3K9me3- enrichment signals at rat OPCs and iOLs surrounding loci of genes which are downregulated and have enhanced OLIG2 binding upon differentiation. **n, o** Heatmap (**n**) and enrichment profiles (**o**) of HA-SETDB1- binding signals at rat OPCs and iOLs surrounding loci of genes which are downregulated and have enhanced OLIG2 binding upon differentiation. Scale bars: 50 μm in (**d**, **e**).

measured by ChIP-seq in primary OPCs and iOLs. SETDB1 occupied at genomic loci encoding genes that are frequently downregulated with enhanced OLIG2 binding upon differentiation (Fig. 2n, o). OLIG2 binding motif was identified by HOMER analysis within HA-SETDB1 peaks among loci of genes down-regulated upon OPC differentiation (Supplementary Fig. 2k). To rule out the possibility that SETDB1 was recruited by other transcription factors, we performed a series of immunoprecipitation assays in rat iOLs. There was weak or undetectable interaction between SETDB1 and these factors, such as NKX2-2, YY1, SOX10 and ZFP191 (Supplementary Fig. 2l). All the evidence indicates that OLIG2 recruits SETDB1 to govern H3K9me3 distribution for gene repression in early stage of OPC differentiation.

**SETDB1 repressor complex is required for oligodendrocyte myelination.** The disintegration of repressive complex by SETDB1 inhibition is equivalent to depriving the role of OLIG2 in transcriptional repression. To investigate the functions of SETDB1 on myelination, *Setdb1* was conditionally deleted in OL lineage by crossing mice harboring floxed Setdb1 alleles and mice carrying *Olig1-Cre*[26,27]. *Setdb1* expression was efficiently eliminated in the oligodendrocyte lineage cells of *Olig1-Cre*; *Setdb1^F/F* mice (Supplementary Fig. 3a). Transgenic mice crossed with *Rosa26-tomato* mice for labelling OPC lineage cells[28]. H3K9me3 level is obviously reduced in the mutant OPC (Supplementary Fig. 3b, c). *Setdb1* mutant mice were born at Mendelian frequencies and were indistinguishable from wildtype (WT) and heterozygous littermates (*Olig1-Cre*; *Setdb1^F/+*) until P10. However, all the mutant mice, but not the WT and heterozygous littermates, developed symptoms associated with myelin-deficiency including tremors, hind limb paralysis, and seizures beginning at P14 (Supplementary Video 1). The mutant mice hardly survived beyond P21 (Fig. 3a). The optic nerve, a myeli-nated white matter, was translucent from mutant mice (Fig. 3b), indicating a severe deficiency in myelin formation. We did notice a measurable hypomyelination in the heterogenous mice (Sup-plementary Fig. 3d). We further examined the ultrastructure of myelin sheath by electron microscopy. In contrast to the large number of myelinated axons in the controls at P14, myelinated axons were hardly detectable in the mutant optic nerve (Fig. 3c, d) and spinal cord (Supplementary Fig. 3e, f). In addition, the expression of myelin gene Plp1 was drastically diminished in the brain (Fig. 3e, f) and spinal cord (Supplementary Fig. 3g, h) of the mutant mice. Consistently, the level of MBP was similarly reduced in the mutant brain (Fig. 3g) and spinal cord (Supple-mentary Fig. 3i). Meanwhile, the differentiated OL marker CC1 was diminished in the brain (Fig. 3h, i) and spinal cord (Sup-plementary Fig. 3j, k) due to *Setdb1* deletion. Noteworthy, Setdb1

inactivation in the OL lineage did not lead to notable alterations in neurons (Supplementary Fig. 4a, d), astrocytes (Supplementary Fig. 4b, d) or microglia (Supplementary Fig. 4c, d). Together, these observations demonstrate the essential role of SETDB1 in myelination.

**SETDB1 is critical for proper myelin repair after demyelina-tion.** Next, the function of SETDB1 was assessed in the remye-lination process following lysolecithin (LPC)-induced demyelination using *Setdb1^Flox/Flox*; *Pdgfra-Cre^ER* mice (*Pdgfra-Cre^ER*; *Setdb1^F/F*) in which SETDB1 was ablated in OPCs upon tamoxifen administration. Local injection of LPC in the white matter induces rapid myelin breakdown followed by regeneration[29]. There are an OPC proliferation and recruitment phase at 7-day post-lesion (Dpl) and then a remyelinating phase at 14 Dpl. LPC was injected into the corpus callosum of *Pdgfra-Cre^ER*; *Setdb1^F/F* mice or control mice (*Pdgfra-Cre^ER*; *Setdb1^F/+*) P60, followed by tamoxifen administration for consecutive 3 days (Fig. 4a). Immunostaining revealed successful creation of lesions (Supplementary Fig. 5a) and that the injured areas were similar between control and mutant mice (Supplementary Fig. 5b). The remyelination in control mice were almost completed indicated by the recurrence of MBP; but significant smaller MBP+ areas were observed in the mutant mice (Fig. 4b). Expression of the myelin-associated genes such as *Plp1* (Fig. 4c, d) and *Mbp* (Supplementary Fig. 5c, d) were decreased in the lesion areas of mutant mice. Ultrastructural analyses on the lesions again revealed markedly lower percentage of remyelinated axons and thinner myelin sheaths in the mutants than that of the controls (Figs. 4e–g). To quantitively evaluate the newly differentiated OLs during myelin regeneration, BrdU was added to label all the proliferative OPCs after LPC injury (Fig. 4a). The total number of CC1+ cell was only slightly declined, but new born OLs (BrdU+ CC1+) was almost vanished in corpus callosum of *Setdb1* mutant mice (Figs. 4h–j). Altogether, our data clear demonstrate that both myelination and remyelination are reliant on the repressive function of OLIG2-SETDB1 complex.

**OLIG2-SETDB1 repressive complex functions at the onset of OPC differentiation.** There are 3 possibilities for the decline in mature oligodendrocytes: i) reduced pool of OPCs; ii) deficiency in OPC differentiation; iii) extensive cell death occurring to the OLs. Both the total number and the proliferation index of PDGFRa+ OPCs were comparable between control and mutant mice (Fig. 5a and Supplementary Fig. 6a). During regeneration, the recruitment of OPCs was not affected by *Setdb1* deletion (Supplementary Fig. 6b, c). Under both myelination and remye-lination circumstance, Enpp6+ (a newly identified marker of

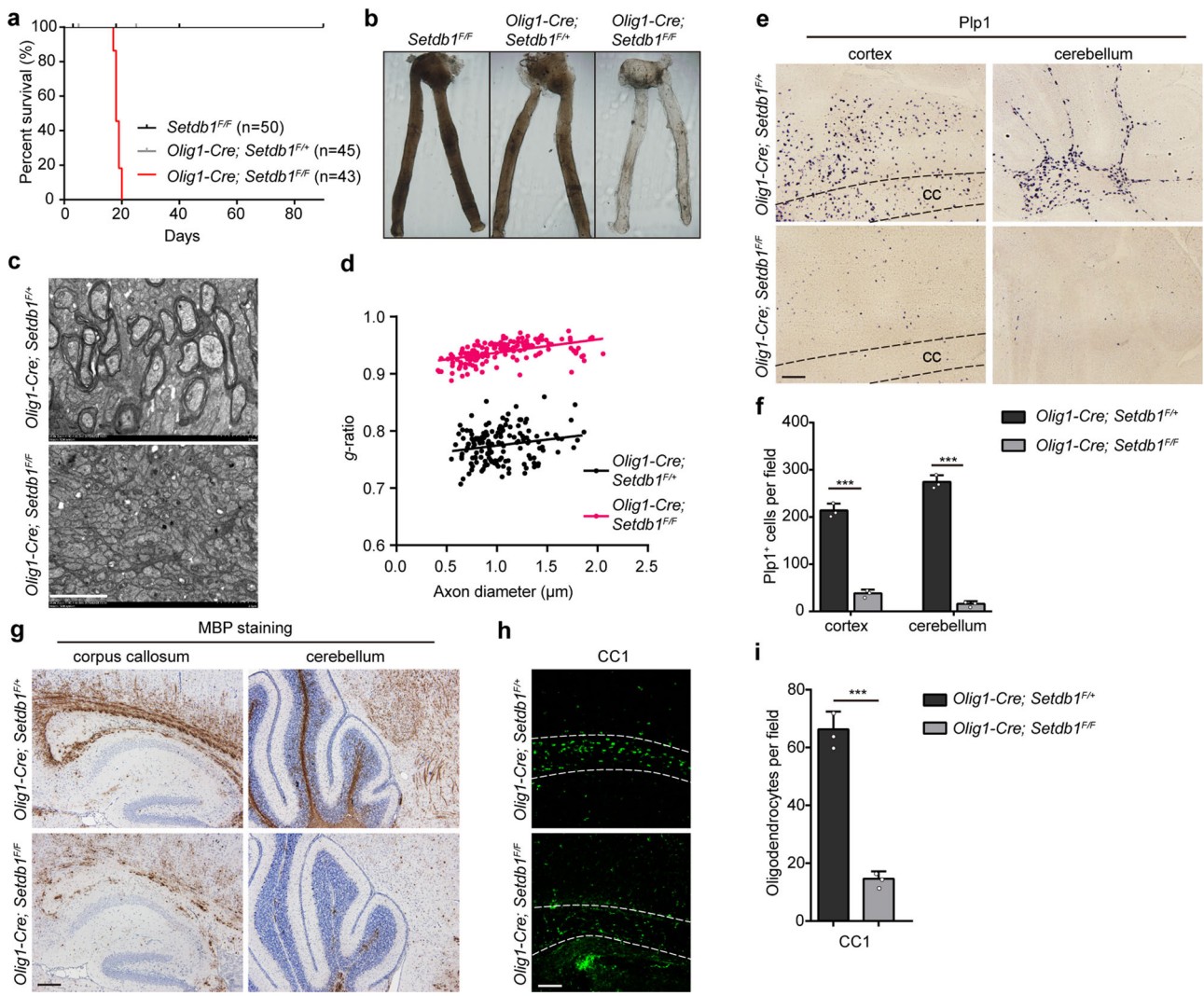

**Fig. 3 SETDB1 is required for oligodendrocyte myelination. a** Survival curve of *Setdb1F/F*, *Olig1-Cre; Setdb1F/+* and *Olig1-Cre; Setdb1F/F* mice. **b** Appearance of optic nerves from *Setdb1F/F*, *Olig1-Cre; Setdb1F/+* and *Olig1-Cre; Setdb1F/F* littermates at P14. **c** Electron micrographs of transverse optic nerve sections from *Olig1-Cre; Setdb1F/+* and *Olig1-Cre; Setdb1F/F* mice at P14. *n* = 3 control and three mutant mice. **d** Quantification of myelin sheath thickness (*g*-ratio) in optic nerves. **e** In situ hybridization of Plp1 in the brain at P14. *n* = 3 control and three mutant mice (**f**) Quantification of Plp1+ cells in the brain white matter. Error bars indicate SEM (cortex, *p* < 0.001; cerebellum, *p* < 0.001. two-tailed unpaired Student's *t* test). **g** Immunolabeling of MBP in the brain at P14. *n* = 3 control and three mutant mice. **h** Immunostaining of CC1 in the corpus callosum at P14. *n* = 3 control and three mutant mice. **i** Quantification of CC1+ cells in the corpus callosum. Error bars indicate SEM (*p* < 0.001. two-tailed unpaired Student's *t* test). Scale bars: 2 µm in (**c**); 100 µm in (**e** and **g**); 50 µm in (**h**).

iOL[30]) cells were evidently reduced in the mutant mice (Fig. 5b, c and Supplementary Fig. 6d, e). The expression of *Enpp6* was declined in vivo and in vitro due to either SETDB1 inhibition or OLIG2-VP64 overexpression (Fig. 5d). Thus, OLIG2-SETDB1 repressive complex is required for iOLs generation.

To clarify the temporal window within which OLIG2-SETDB1 complex functions, various mouse models were generated to interrogate the repressive complex in different OL lineages. When the complex was disassembled in the OPCs of *NG2-Cre; Setdb1F/F* mice, myelination was substantially reduced in the cortex (Fig. 5e) and spinal cord (Supplementary Fig. 6f); and the abundance of CC1+ cells was diminished in the brain and spinal cord (Fig. 5f and Supplementary Fig. 6g, h). To disassemble the complex after iOLs generation, we generated *Cnp-Cre; Setdb1F/F* mice (Fig. 5g). These mutant mice survived indistinguishably as the control littermates (Fig. 5h). The number of Plp1+ cell was comparable between mutant and control mice (Supplementary Fig. 6i, j). Ultrastructure analyses of the optic nerves revealed that

myelination and *g*-ratio of axons were also similar (Fig. 5i and Supplementary Fig. 6k). Consistently, MBP and CC1 immunostaining showed that mutant mice did not exhibit notable alternations in myelin (Supplementary Fig. 6l) and OLs at P15 (Fig. 5j and Supplementary Fig. 6m) and P60 (Supplementary Fig. 6n).

The white/grey matter of spinal cord and brain are sequentially myelinated[31]. For example, myelinogenesis has been accomplished in the white matter of spinal cord but not the grey matter of spinal cord nor the brain by P3[32]. We took advantage of *Sox10-CreER* that permits simultaneous tamoxifen-inducible targeting of SETDB1 in the OPCs, iOLs and OLs. When the mice were treated with tamoxifen at P3 (Fig. 5k), striking hypomyelination was observed in the brain and the gray matter but not the white matter of spinal cord. MBP intensity and the number of CC1+ cells in the cortices of P15 mutant mice were largely reduced compared to the controls (Fig. 5l, m). In the spinal cord, these phenotypes were only observed in the gray matter but not white

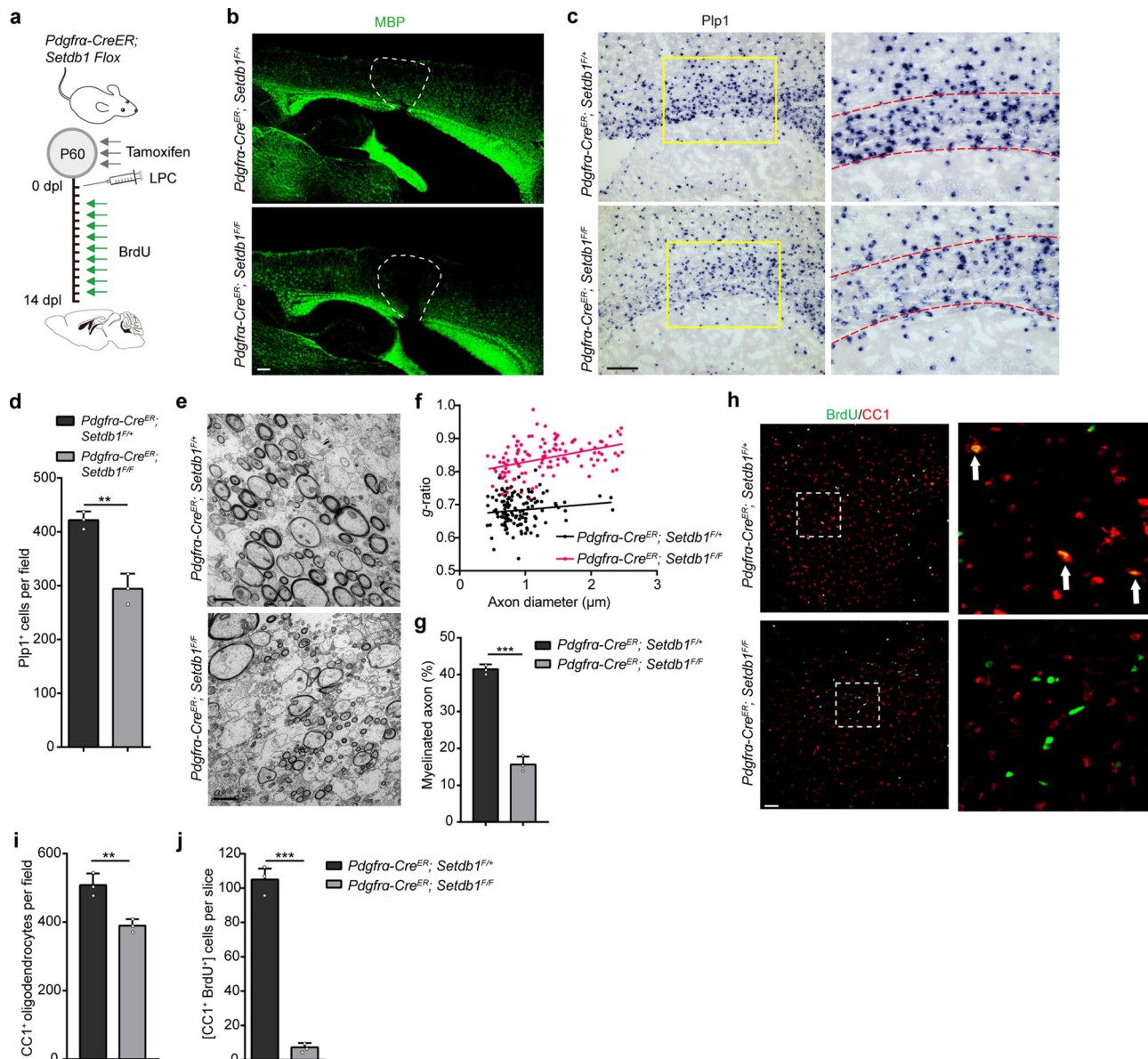

**Fig. 4 SETDB1 is critical for proper myelin repair after demyelination. a** Schematic diagram showing tamoxifen administration to *Pdgfra-Cre^ER*; *Setdb1 Flox* mice in 3 consecutive days at P60, followed by LPC injection and BrdU administration. Brain tissue were collected at 14 Dpl. **b** Immunolabeling of MBP in the LPC lesions of *Pdgfra-Cre^ER*; *Setdb1^F/+* and *Pdgfra-Cre^ER*; *Setdb1^F/F* mice at 14 Dpl. *n* = 3 control and three mutant mice. **c** In situ hybridization of Plp1 in the lesions at 14 Dpl. Boxed areas in left panels are shown at a high magnification in corresponding right panels. *n* = 3 control and three mutant mice. **d** Quantification of Plp1^+ cells in the LPC lesions. Error bars indicate SEM ($p$ = 0.0023. two-tailed unpaired Student's *t* test). **e** Electron micrographs in corpus callosum of lesions from *Pdgfra-Cre^ER*; *Setdb1^F/+* and *Pdgfra-Cre^ER*; *Setdb1^F/F* mice at 14 Dpl. *n* = 3 control and three mutant mice. **f** Scatterplot of *g*-ratio at 14 Dpl in lesions area. **g** Percentage of remyelinated axons at 14 Dpl in lesions area. Error bars indicate SEM ($p$ < 0.001. two-tailed unpaired Student's *t* test). **h** Immunostaining for BrdU and CC1 in LPC lesions in corpus callosum from *Pdgfra-Cre^ER*; *Setdb1^F/+* and *Pdgfra-Cre^ER*; *Setdb1^F/F* mice at 14 Dpl. Boxed areas in left panels are shown at a high magnification in corresponding right panels. *n* = 3 control and three mutant mice. **i, j** Quantification of CC1^+ cells (**i**) and CC1^+/BrdU^+ cells (**j**) in the LPC lesions area. Error bars indicate SEM (**i**, $p$ = 0.0061; **j**, $p$ < 0.001. two-tailed unpaired Student's *t* test). Scale bars: 100 μm in (**b**, **c**); 2 μm in (**e**); 50 μm in (**h**).

matter (Fig. 5n–p). The turnover rate of myelin in the adult mice is low and most of OPCs are dormant[33]. Depleting *Setdb1* in the OLs at P60 had no notable impact on the myelination throughout our experimental timeframe (up to P90) (Supplementary Fig. 6o-q). TdT-mediated dUTP Nick-End Labeling (TUNEL) assay failed to detect obvious cell death in *Setdb1* mutant brains (Supplementary Fig. 6r). Together, these different mouse models demonstrate a temporary function of SETDB1 repressive complex in the early but not late stage of OPC differentiation and that SETDB1 is indispensable for the maintenance of myelin.

**OLIG2 recruits SETDB1 to silence inhibitors of oligodendrocyte differentiation**. Then we performed H3K9me3 ChIP-seq in control and *Setdb1* mutant iOLs prepared from mouse brain. H3K9me3 peak density was most diminished around those gene loci occupied by OLIG2 in the iOLs from mutant mice (Supplementary Fig. 7a). Inhibition of SETDB1 and overexpression of OLIG2-VP64 caused similar upregulated DEGs, as revealed by GO analysis (Supplementary Fig. 7b, c). Pro-myelinogenesis genes including *Enpp6, Sox10, Plp1, Mbp, Cnp* and *Mag* were down-regulated as expected in two transcriptomes (Fig. 6a, b). Sox11[34]

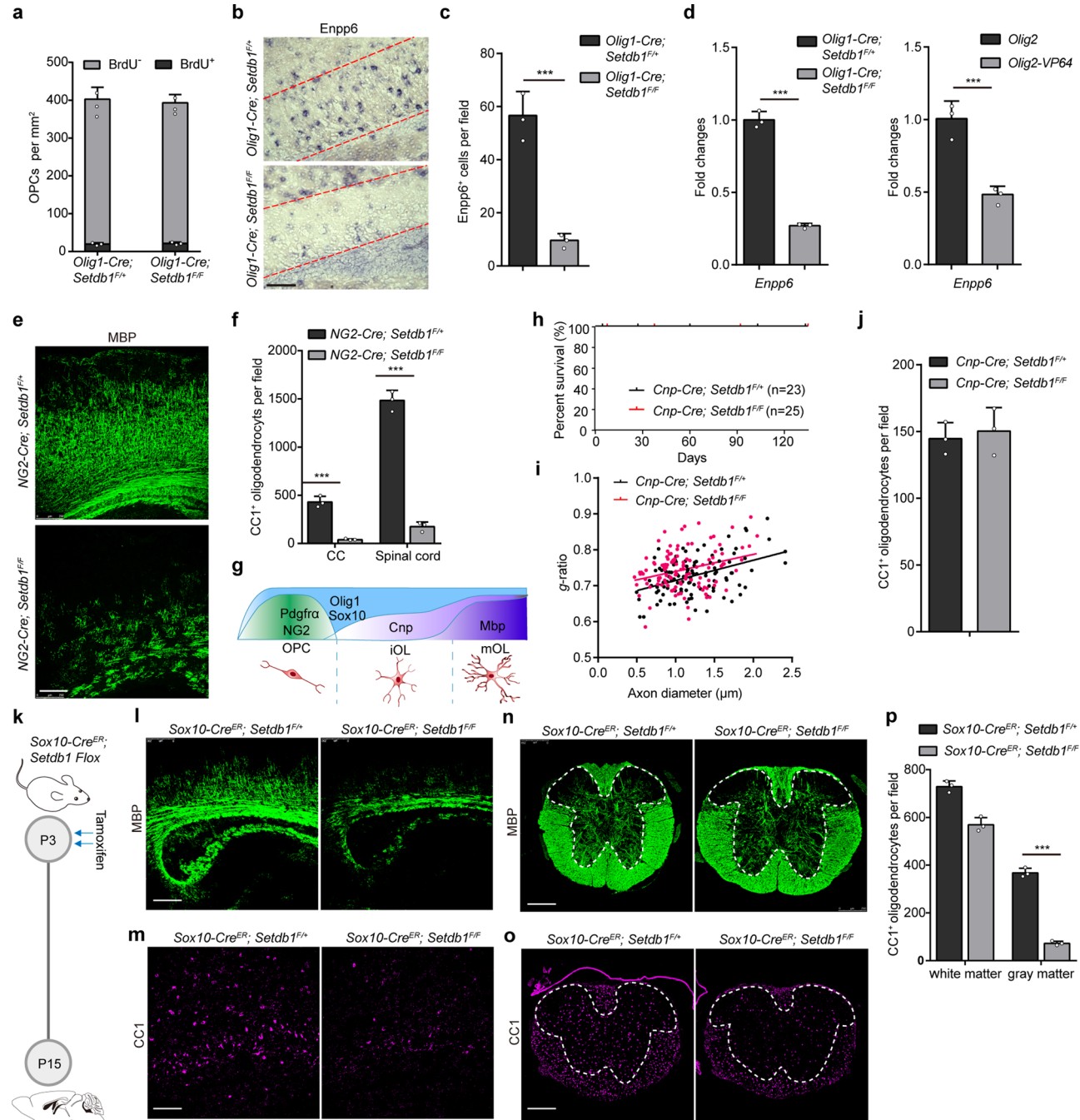

**Fig. 5 OLIG2-SETDB1 repressor complex functions at the onset of mouse OPC differentiation. a** Quantification of BrdU[+] and PDGFRα[+] cells in the corpus callosum of *Olig1-Cre; Setdb1[F/+]* and *Olig1-Cre; Setdb1[F/F]* mice at P14. **b** In situ hybridization of Enpp6[+] in the corpus callosum at P14. *n* = 3 control and three mutant mice. **c** Quantification of Enpp6[+] cells in the corpus callosum at P14. Error bars indicate SEM (*p* < 0.001. two-tailed unpaired Student's *t* test). **d** Expression of Enpp6 in spinal cords of Setdb1 control and mutant mice (left panel) and rat iOLs transfected with *Olig2* or *Olig2-VP64* (right panel). Error bars indicate SEM (*p* < 0.001. two-tailed unpaired Student's *t* test). **e** Immunostaining of MBP in the corpus callosum from *NG2-Cre; Setdb1[F/+]* and *NG2-Cre; Setdb1[F/F]* mice at P14. *n* = 3 control and three mutant mice. **f** Quantification of CC1[+] cell in the corpus callosum and spinal cord from *NG2-Cre; Setdb1[F/+]* and *NG2-Cre; Setdb1[F/F]* mice at P14. Error bars indicate SEM (*p* < 0.001. two-tailed unpaired Student's *t* test). **g** Schematic diagram showing markers of each stage in oligodendrocyte lineage. **h** Survival curve of *Cnp-Cre; Setdb1[F/+]* and *Cnp-Cre; Setdb1[F/F]* mice. **i** Quantification of myelin sheath thickness (*g*-ratio) in optic nerves from *Cnp-Cre; Setdb1[F/+]* and *Cnp-Cre; Setdb1[F/F]* mice at P14. *n* = 3 control and three mutant mice. **j** Quantification of CC1[+] cells in the spinal cord of *Cnp-Cre; Setdb1[F/+]* and *Cnp-Cre; Setdb1[F/F]* mice at P14. *n* = 3 control and three mutant mice. **k** Schematic diagram showing tamoxifen administration to *Sox10-Cre[ER]; Setdb1 Flox* mice in two consecutive days at P3, followed by tissue collection at P15. *n* = 3 control and three mutant mice. **l–o** Immunostaining of MBP and CC1 in the corpus callosum (**l**, **m**) and spinal cord (**n**, **o**) from *Sox10-Cre[ER]; Setdb1[F/+]* and *Sox10-Cre[ER]; Setdb1[F/F]* mice at P14. *n* = 3 control and three mutant mice. **p** Quantification of CC1[+] cells in the white matter or gray matter of spinal cord. Error bars indicate SEM (*p* < 0.001. two-tailed unpaired Student's *t* test). Scale bars: 100 μm in (**b**); 250 μm in (**e** and **l–o**).

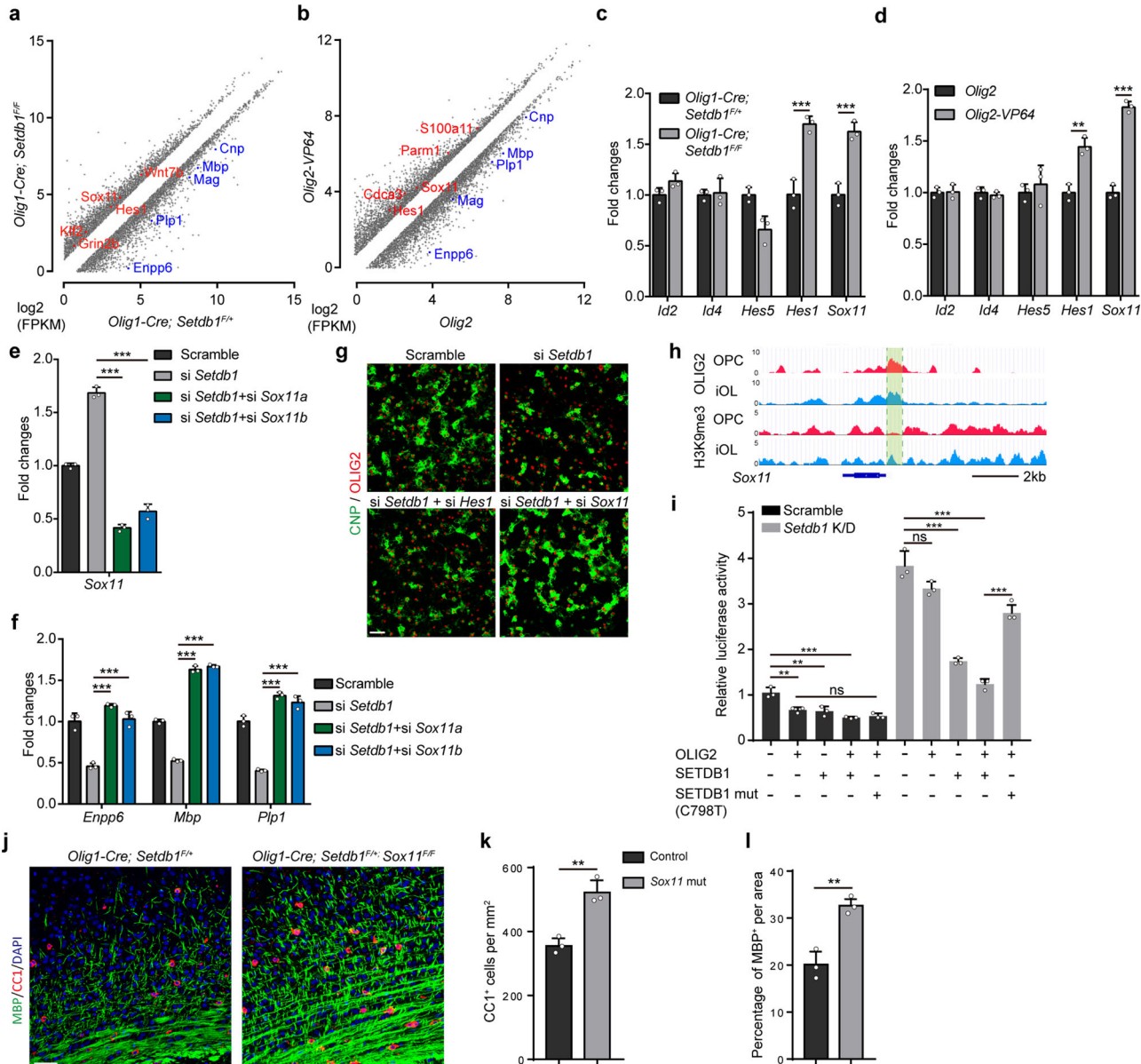

**Fig. 6 Olig2 recruits SETDB to silent inhibitors of oligodendrocyte differentiation. a, b** Scatterplot showing differentially expressed transcripts between iOL from *Olig1-Cre; Setdb1*^F/+ and *Olig1-Cre; Setdb1*^F/F brain cortex (**a**) and between rat OPCs transfected with *Olig2* and *Olig2-VP64* under differentiation condition (**b**). **c, d** Expression of well-known inhibition factors to rat OPC differentiation. Error bars indicate SEM (***$p < 0.001$; Hes1 in (**d**), $p = 0.0029$. two-tailed unpaired Student's *t* test). **e** Efficiency of Sox11 knockdown in rat OPCs. Error bars indicate SEM (***$p < 0.001$; Hes1 in Olig2-VP64, $p = 0.0029$. one-way ANOVA). **f** Expression of *Enpp6*, *Mbp* and *Plp1* was measured in rat OPCs under differentiation condition following transfection with indicated siRNA. Error bars indicate SEM ($p < 0.001$. one-way ANOVA). **g** Immunolabeling of CNP and OLIG2 in OPCs under differentiation condition following transfection with indicated siRNA. $n = 3$ independent experiments. **h** Genome browser visualization of OLIG2 or H3K9me3-targeting regions on the gene loci of *Sox11* during rat OPC differentiation. **i** Luciferase activity of Sox11 promoter-driven reporters in control (Scramble) or Setdb1 knockdown (Setdb1 K/D) 293T cells transfected with indicated plasmids. Error bars indicate SEM (***$p < 0.001$; OLIG2+ in Scramble, $p = 0.0078$; SETDB1+ in Scramble, $p = 0.0095$. one-way ANOVA). **j** Immunolabeling for MBP and CC1 on the corpus callosum from *Olig1-Cre; Setdb1*^F/+ and *Olig1-Cre; Setdb1*^F/+; *Sox11*^F/F mice at P14. $n = 3$ control and three mutant mice. **k, l** Quantification of CC1+ oligodendrocytes density (**k**) and the percentage of MBP+ area (**l**) in corpus callosum of control and Sox11 mutant mice. Error bars indicate SEM (**k**, $p = 0.0064$; **l**, $p = 0.0052$. two-tailed unpaired Student's *t* test). Scale bars: 50 μm in (**g**, **j**).

and Hes1[35] (Supplementary Fig. 7d) are two inhibitory transcriptional factors for myelin. Their levels were increased in two transcriptomes (Fig. 6a, b), which was not observed for other well-known transcriptional factors with anti-myelinogenesis properties (Fig. 6c, d). However, only Sox11 but not Hes1 silencing restored the capacity of myelinogenesis in the *Setdb1*-deficient rat OPCs (Fig. 6e–g and Supplementary Fig. 7e, f). *Sox11* loci were occupied with OLIG2 and modified with H3K9me3 (Fig. 6h), and H3K9me3

density was reduced in the promoter region of *Sox11* in the *Setdb1* mutant iOLs (Supplementary Fig. 7g). ChIP-qPCR assay was performed to confirm the decreased H3K9me3 enrichment when OLIG2 was knocked down in the rat iOLs (Supplementary Fig. 7h). The regulation of *Sox11* by OLIG2-SETDB1 complex was verified by luciferase assays. To avoid potential interference from endogenous SETDB1, *we generated Setdb1*-deficient HEK293T cells (Supplementary Fig. 7i). OLIG2 inhibited *Sox11*

transcription with the exogenous SETDB1. In contrast, the enzy-matic dead SETDB1 mutant (C798T)[36] displayed almost no repressive effect on *Sox11* expression (Fig. 6i). Next, we mutated two conservative OLIG2 binding motif in the promoter region of *Sox11*, OLIG2-SETDB1 complex is less potent to suppress *Sox11* transcription (Supplementary Fig. 7j). To explore if the regulatory relationship between OLIG2-SETDB1 complex and SOX11 exits in vivo, we injected the *Setdb1* mutant mice with ASO (modified antisense oligonucleotides) against Sox11 mRNA at P6, and brain tissues were collected at P14 (Supplementary Fig. 7k). Cy3-tagged ASO showed high efficiency in the forebrain (Supplementary Fig. 7l), and *Sox11* was efficiently knocked down by ASO (Sup-plementary Fig. 7m). The MBP intensity was remarkably increased in the corpus callosum of Sox11 ASO injected *Setdb1* mutant mice (Supplementary Fig. 7n) *Sox11* is widely expressed in the neurons as well. To exclude the influence by neurons with deficient *Sox11*, OL-specific *Sox11* knockout mice were generated. For unknown reasons, the *Setdb1/Sox11* double mutant mice exhibited severe developmental defects. Nevertheless, the number of CC1+ OLs and the myelinated fibers was significantly increased in *Olig1-Cre; Setdb1^{F/+}; Sox11^{F/F}* mice compared with that in *Setdb1* hetero-zygous mice (Fig. 6j-l). Myelination defects caused by *Setdb1* deficiency was remarkably rescued by suppressing *Sox11*, high-lighting the importance of the OLIG2-SETDB1-SOX11 axis in the myelinogenesis.

## Discussion

Master regulators has been discovered to govern the proper dif-ferentiation of stem cells in many tissues. Epigenetic factors, working as the executors to modulate the transcriptional machinery, were often recruited by master regulators to fulfill their functions. In our study, we show that OLIG2, a master regulator of oligodendrocyte specification, functions as a tran-scriptional repressor beyond its canonical role as an activator. By recruiting SETDB, OLIG2 builds the H3K9me3 modification on the genomic loci, such as *Sox11*, destined to be downregulated for OPC differentiation.

Master regulators play multiple roles in tissue specification. When tissue specification starts the transcriptomic network entails orchestration with certain genes activated, while others repressed. In these tightly-regulated processes, often one or two most essential transcriptional factors (known as master regulator) function to control all the downstream transcriptional machinery. Increasing evidence supported that master regulators are capable of governing the differentiation process via distinct mechanisms. They can be both activators and repressors, which is largely dependent on the identity of cofactors they cooperate with. This is well exemplified by MyoD in myogenesis, RUNX2 in osteoblast genesis and YY1 in early embryogenesis[17,37,38].

The repressor role of OLIG2 is important and essential. The expression of OLIG2 is highly specific to the oligodendrocyte and motor neurons. Early studies on motor neuron development show that activator form of OLIG2 (OLIG2-VP16) inhibits motor neuron generation[15,39]. But the molecular mechanisms are not clear. Meanwhile, OLIG2 has been found to be a master regulator in oligodendrocyte development[26,40]. Whereas OLIG2 has been reported to be an essential activator in OPC differentiation by recruiting SMARCA4, an ATP-dependent chromatin remodeler[7], we find that OLIG2 may have a repressive function in OPC dif-ferentiation by analyzing several public sequencing database[7,20]. Furthermore, we reveal that SETDB1, recruited by OLIG2, function as a major executor to repress a set of genes destined to be downregulated upon OPC differentiation.

Changes of epigenetic status affect the oligodendrocyte devel-opment. In the field of oligodendrocyte, several epigenetic factors

were reported to be essential for myelination. These include DNA methylation modifiers[41], RNA modifiers[42,43], chromatin remodelers[7,44,45], histone acetylation enzymes[46] and histone methylation enzymes[24,47,48]. Most of these modifiers or enzymes are well-studied in the context of OPC differentiation, while H3K9me3 modifier SETDB1 remains not. Casaccia and collea-gues did a study on the impact of H3K9me3 modification on OPC development in vitro[24]. We noticed that their conclusions are contradictory to ours, which might be attributed to differences in the methods to deliver shRNA. In their study, lentivirus was used to delivery shRNA. However, we used electroporation. Moreover, they did not investigate SETDB1, the leading actor in our study, in OPC development. And we showed that SETDB1, a key H3K9me3 modifier, promote OPC differentiation by repressing inhibitory effector *Sox11*.

SETDB1 and H3K9me3 play various roles in multiple tis-sues. SETDB1 is expressed in almost all cell types in mammals, and various functions have been established for SETDB1 in different tissues. In the embryonic stem cells, SETDB1 represses the differentiation-associated genes to maintain the pluripotency. And repression of chimeric transcripts may also contribute to the process[49]. On the other hand, SETDB1-dependent repression of endogenous retrovirus is crucial to proper development of various pluripotent stem cells, includ-ing primordial germ cells, T cells and gut stem cells. In most of these cases, aberrant activation of endogenous retrovirus resulted from *Setdb1* deficiency leads to cell death[19,50,51]. Interestingly, cell death is not detectable in the *Setdb1* knockout cells in this current study. Therefore, the roles of SETDB1 are variable, which is dependent on different master regulators and microenvironment.

## Methods

All research performed complies with all relevant ethical regulations. All animals were maintained in the core animal facility at Xiamen University, and all experi-mental procedures involved were performed according to protocols approved by the Institutional Animal Care and Use Committee at Xiamen University. The mouse strains used in this study were generated and maintained on a mixed C57Bl/6; 129 Sv background. Mice analyzed were litter mates and sex-matched whenever possible. Mice and rats were housed in a vivarium with 12 h light /dark cycle with free access to water and food. Male and female animals were used for all experiments.

**Contact for reagent and resource sharing**. Further information and requests for resources and reagents should be directed to and will be fulfilled by the Lead Contact, Wei Mo (wmo@xmu.edu.cn).

**Animals**. The following transgenic mice strains were used:

*Pdgfra-Cre^{ER}* and *Sox10-Cre^{ER}* mice were generously provided by Dr. William D. Richardson (Wolfson Institute for Biomedical Research, University College London, London, UK). *Cnp-Cre* (generated by Dr. Klaus Nave), *Olig2^{Flox/Fox}* and *Sox1^{Flox/Flox}* mice were generously provided by Dr. Luis F. Parada (Department of Developmental Biology, University of Texas Southwestern Medical School, Dallas, TX, USA). *NG2-Cre* and *Olig1-Cre* mice were from Jackson Laboratory (stock number: 008533 and 011105). *Setdb1^{Flox/Flox}* mice were generously provided by Dr. Yan Jiang (Institutes of Brain Science, State Key Laboratory of Medical Neurobiology and MOE Frontier Center for Brain Science, Fudan University, 200032, Shanghai, China).

**Quantitative RT-PCR and primers used**. Total RNA from animal tissues and cells were isolated by using Trizol reagent according to the manufacturer's instructions (Invitrogen, Cat# 15596026). Reverse transcription was performed using the GoScript™ Reverse Transcription System (Promega, Cat# A2791). Then cDNA was amplified by real-time quantitative RT-PCR using SYBR Green (Vazyme, Cat# Q121-02) reagent. Primer sequences for mouse genes include: Id2-f, cggtgaggtccgttaggaaaa and Id2-r, gcttggagtagcagtcgttca; Id4-f, atcccgcccaacaagaaagt and Id4-r, atgctgtcaccctgcttgtt; Hes1-f, aagaggcgaagggcaagaata and Hes1-r, ggaggtgcttcacagtcatttc; Hes5-f, atgctcagtcccaaggagaaa and Hes5-r, cgaaggctttgctgtgtttca; Sox11-f, agatcgagcgcaggaagatca and Sox11-r, agtcggga-taatcagccatgtg; Enpp6-f, cattttggatgaacagcacgg and Enpp6-r, cacggatctgattggagcag; Gapdh-f, ccactcacggcaaattcaac and Gapdh-r, ctccacgacatactcagcac. Primer sequences for rat genes include: Id2-f, acagaaccaaacgtccagg and Id2-r, ggaaaaagtccccaaatgcc; Id4-f, gctcaacactgacccgg and Id4-r, tcttaatttctgctctggccc;

Hes1-f, agaaaaattcctcgtccccg and Hes1-r, tttcatttattcttgcccggc; Hes5-f, accagcc-caactccaaac and Hes5-r, agtaaccctcgctgtagtcc; Sox11-f, acttcgagttccccgactac and Sox11-r, catcctcttttatcctgaccgc; Enpp6-f, agtggattcaggaacgaggc and Enpp6-r, gtga-tatataacgggcccttgg; Mag-f, tcaacagtccctaccccaag and Mag-r, gagaagcagggtgcagtttc; Mbp-f, aaatcggctcacaagggattc and Mbp-r, ctcccagcttaaagattttggaaa; Plp1-f, ggcgactacaagaccaccat and Plp1-r, cctagccattttcccaaaca; Setdb1-f, ggtcagaagga-gagtgaactg and Setdb1-r, ttgttcttgggtgtctcttcg; Ehmt1-f, ctctaaattcctcgcttcctgg and Ehmt1-r, gctttctgtcctctgtgtctcg; Ehmt2-f, tccgagaactgtgaaacgtc and Ehmt2-r, ccgtccttatcataccagcatc; Suv39h1-f, aaaccgtgtagtccagaaagg and Suv39h1-r, tcccaca-tactccataacaaagc; Gapdh-f, tccagtatgactctacccacg and Gapdh-r, cacgacatactcagccaccag.

**Cells and cell cultures**. Primary mouse OPCs were isolated from cortices of P4-P8 mouse brains. Cerebral cortexes were dissected and then dissociated by mechanical trituration until the cell suspension has no small clumps. Then dissociated mouse cortices cells were resuspended in panning buffer. Suspension was sequentially immunopanned on anti-GalC and anti-O4 antibody coated plates. The adherent OPCs were trypsinized and plated onto poly-D-lysine coated plates. The cultures were maintained under proliferating condition which is DMEM/F-12 (GIBCO, Cat#11330032) with addition of 2% B-27 (GIBCO, Cat#A3353501), 1% N2 (GIBCO, Cat#A1370701), 20 ng/ml PDGF-AA (Peprotech, Cat#100-13 A), 10 ng/ml CNTF (Peprotech, Cat#450-13), 20 ng/ml bFGF (Sino Biological, Cat#10014-HNAE), and 1 ng/ml NT3 (Peprotech, Cat#450-03). For OPCs differentiation, 60 nM L-triiodothyronine (T3; Sigma-Aldrich, Cat#T6397) was added to the media and PDGF-AA, bFGF and NT3 were removed. For iOLs, we harvest cells at 24 h after T3 induction. And for mOLs, we induce OPCs differentiation for 72 h. Rat OPCs were obtained from P2–P4 rat brains as mouse OPCs with slight modifications. Briefly, dissociated rat cortices cells were sequentially immunopanned on anti-GalC and anti-A2B5 antibody coated plates to harvest OPCs. For electroporation we used a Lonza Nucleofector device. And OPCs were induced to differentiate 36 h later. The culture condition is the same as mouse OPCs

HEK293T cells were cultured in DMEM (GIBCO, Cat# 12430047) supplemented with 10% FBS (GIBCO, Cat#16000044) and 1× penicillin-streptomycin (GIBCO, Cat#15140148) and maintained at 37 °C in humidified air containing 5% $CO_2$. HEK293T Transfections were performed using Polyetherimide (fushen biotechnology, Cat#FSF0001).

**Retrovirus injection**. After *Olig2^{Flox/Flox}* mice at P5 were anesthetized, the skin above the skull was disinfected and a small cut was made. Injection of the virus into the corpus callosum was performed using the following coordinates from Bregma: anteroposterior 0.2, mediolateral ±0.5, dorsoventral −1.6 as described[52]. Subsequently, the wound was sutured with silk and xylocaine was applied locally. After the surgery, mice were returned to their home cages with parents.

**Western blotting**. Cultured cells or mouse brain tissues were homogenized in chilled RIPA lysis buffer (Thermo Fisher Scientific, Cat#89900) for 3 min with homogenate machine, and subsequently centrifuged at 13,000 g for 10 min at 4 °C centrifugal machine, and transferred to a clean 1.5 ml tube. Sample containing 3 g brain tissues were homogenized in chilled RIPA lysis buffer (Thermo Fisher Scientific, Cat#89900) for 3 min with homogenizer machine. Protein was transferred into PVDF membrane (EMD-Millipore, Cat#IPVH00010) after running SDS-PAGE gels in an ice-cold buffer. Immunoblots were probed with indicated antibodies. Primary antibodies include: rabbit anti-SETDB1 (Proteintech, 1:3000, Cat#11231-1-AP), rabbit anti-OLIG2 (talent biomedical, 1:3000, Cat#AP0337), mouse anti-HA probe (Santa Cruz Biotechnology, 1:10000, Cat#SC-7392), rabbit anti-PDGFRα (Santa Cruz Biotechnology, 1:3000, Cat#sc338), mouse anti-GFAP (talent biomedical, 1:3000, Cat#AM0123), rabbit anti-IBA1 (WAKO, 1:3000, Cat#019-19741), mouse anti-NeuN (EMD Millipore, 1:3000, Cat#mab377), mouse anti-CNP (EMD Millipore, 1:3000, Cat#MAB326), goat anti-MBP (Santa Cruz Biotechnology, 1:3000, Cat#sc-13914), mouse anti-GAPDH (Santa Cruz Biotechnology, 1:3000, Cat#365062). Secondary antibodies include: Goat anti-mouse (Thermo Fisher Scientific, 1:5000, Cat#G-21040), goat anti-rabbit (Thermo Fisher Scientific, 1:5000, Cat#G-21234), donkey anti-goat (Thermo Fisher Scientific, 1:5000, Cat#A16005).

**Co-immunoprecipitation**. Brain or cell lysates were pre-cleared with protein A-Sepharose beads (YEASEN Biotech, Cat#36403ES25). After washing, lysates were incubated with primary antibody for 8 hr at 4 °C. After incubation, complexes were precipitated with 10 μl of protein A-Sepharose beads with gentle agitation at 4 °C for 2 h; nonspecific IgG was used as a negative control. Beads were washed, and the immunoprecipitated protein complex was loaded onto SDS-PAGE gels and processed for western blotting.

**Mass spectrometry**. Brain lysates were pre-cleared with protein A-Sepharose beads (YEASEN Biotech, Cat#36403ES25) and then incubated with primary antibody and protein A-coated Sepharose beads for 2 hr at 4 °C. Following three washes in lysis buffer, bound proteins were eluted from the beads in SDS sample buffer. For analysis by mass spectrometry, immunoprecipitates were first resolved

by SDS-PAGE, with each lane cut into four slices for in-gel digestion. Tryptic peptides were analyzed by liquid chromatog-raphy-tandem mass spectrometry (LC-MS/MS, Thermo Orbitrap Fusion Lumos). Basing on the unique peptide numbers of proteins detected, we made the scatter plot of proteins bound to OLIG2.

**Tissue and immunohistochemistry**. Mice were anesthetized before sacrifice and then perfused with ice-cold phosphate-buffered saline (PBS) followed by 4% paraformaldehyde (PFA). The brain and spinal cord were dissected, fixed in 2% PFA overnight, dehydrated in 25% sucrose at 4 °C, embedded in OCT (Leica, 14020108926) and processed for cryo-sections at 12 μm. Cultured cells were briefly fixed with a chilled 4% PFA/PBS solution for 10 min and permeabilized with 0.4% Triton X-100/PBS for 10 min on ice. For immunohistochemistry, cryo-sections were permeabilized and blocked in blocking buffer (0.4% Triton X-100 and 3% BSA in PBS) for 30 min at room temperature (RT) and overlaid with diluted primary antibodies overnight at 4 °C. After washing with PBS, sections were incubated with secondary antibodies conjugated to Cy2, Cy3 and Cy5 (Jackson ImmunoResearch Laboratories, 1:1000) and DAPI (Thermo Fisher Scientific, Cat#D1306) for 1 h at RT, washed three times in PBS and then mounted with Mounting medium. For Immunocytochemistry, Cultured cells were briefly fixed with a 4% PFA/PBS solution for 10 min at RT, permeabilized and blocked by the pre-cold buffer (0.4% Triton X-100 and 3% BSA in PBS) at 4 °C. Coverslips were covered with diluted primary antibodies overnight at 4 °C. Secondary antibodies and DAPI were incubated for 1 h at RT. Immuno-fluorescence images were taken using Leica SP8 and Zeiss LSM 780 confocal laser microscope.

**In situ hybridization**. In situ hybridization was performed on fresh frozen sagittal brain sections (18 μm), using digoxigenin (DIG) -labeled antisense probes for murine *Plp1/Dm-20*, *Mbp* and *Enpp6* as described previously[30,53,54]. The probe was synthesized by T3 RNA polymerase (Promega, Cat#P208C) and labeled with DIG RNA label mix (Roche, Cat#11277073910). An anti-DIG antibody conjugated with alkaline phosphatase (Roche, Cat#11093274910) was used to probe sections and followed by stained with 5-bromo-4-cloro-3-indolyl phosphate (BCIP) (Solarbio, Cat#B8090) /nitro blue tetrazolium (NBT) (Solarbio, Cat#N8140) chromogenic substrates.

**Electron microscopy**. Tissue processing was performed essentially as described previously[7]. Briefly, mice were deeply anesthetized, perfused with pre-cold sodium cacodylate buffer. Optic nerve and spinal cord were dissected immediately and postfix with 2.5% glutaraldehyde overnight at 4 °C. Treated with 1% osmium tetroxide, dehydrated, and embedded into PolyBed resin. 70-nm ultrathin sections stained with lead citrate for electron microscopy imaging by using JEM-2100HC and Hitachi HT-7800.

**Lysolecithin-induced demyelinating injury**. Lysolecithin (LPC)-induced demyelination was carried out in the corpus callosum (CC) and of 8-week-old *Pdgfra-Cre^{ER}*; *Setdb1 Flox* mice as described. Briefly, for CC injury, after exposing the skull, 2 μl of 1% (W/V) LPC (Sigma, Cat#L0906) in PBS was stereotactic injected into CC. The coordinates were: 1 mm backward to bregma, 1 mm lateral to bregma, and 2 mm deep relative to skull surface. Brain tissue carrying the lesions were harvested at different time points.

**Tamoxifen administration**. Tamoxifen (sigma, Cat# T5648) was dissolved in corn oil at a concentration of 20 mg/ml and stored at −20 °C. Neonatal mice were administrated with tamoxifen through subcutaneous injection in the back of pups at a dosage 40 μg/g (gram, body weight) at a single injection daily for two stitutive days from P3-P4. Young adult mice acquired daily tamoxifen injection through i.p. injection at a dosage of 125 μg/g (gram, body weight) at a single injection daily for three constitutive days.

**BrdU assay**. For the proliferation assay, mice were pulsed with 100 μg/g (gram, body weight) of BrdU at 2 h before sacrificed. For differentiation assay, BrdU was administrated into drinking water with 1 mg/ml for 14 days.

**Luciferase reporter assay**. HEK 293 T Cells were transfected with pGL3-Hes1-Luc or pGL3-Sox11-Luc reporter constructs, β-galactosidase (β-gal) expression vectors and other relevant plasmids. The promoter of Sox11 we cloned is -1kb to 0 (TSS), where Olig2 binds as shown in ChIP-seq result. At 36 h post transfection, cells were harvested and luciferase activities were measured in triplicate and normalized to internal β-gal activity to normalize for transfection efficiency.

**Sox11-targeting ASO design**. Second generation ASO was designed to target mouse Sox11. ASO consisted of 18-mer nucleotide sequences (5′-tttatttactctattggcc-3′) with 2′-O-methoxyethyl (MOE) modifications and a mixed backbone of phosphorothioate and phosphodiester internucelotide linkages as previously

described[55]. ASOs that efficiently reduced Sox11 mRNA were selected for in vivo screening and tolerability studies.

**RNA-seq and data analysis**. Total RNA from purified iOLs were prepared using Illumina RNA-Seq Preparation Kit and subjected to 150-bp double-end sequencing with an Illumina sequencer as previously described[56]. At least 20 million clean reads of sequencing depth were obtained for each sample. RNA-seq raw data were initially filtered to obtain clean data after quality control. Clean data were aligned to the mouse genome (mm10) or rat genome (rn6) by HISAT2[57]. Raw counts for each gene were calculated by Htseq[58]. StringTie was used to estimate the expression level of detected genes[59]. DEGs were defined as genes with FDR less than 0.001 and fold change larger than 1.5.

**Chromatin-immunoprecipitation followed by sequencing (ChIP-seq)**. ChIP-seq in rat OPCs and iOLs, and mouse iOLs were performed as previously described[7], with slight modifications. Briefly, for ChIP-seq, approximately $5 \times 10^6$ cells were cross-linked with 1% formaldehyde for 10 min at room temperature and then were quenched with 125 mM glycine for 5 min. Sonicated chromatin was used for immunoprecipitation by incubation with 4 µg appropriate antibodies (anti-OLIG2, Millipore, cat# ab9610; anti-H3K9me3, Active Motif, cat# 39161; anti-HA tag, Sant Cruz, cat# SC-7392) overnight at 4 °C. Immunoprecipitated complexes were collected using 40 µl protein A plus agarose beads (Millipore, cat#16-156). Subsequently, beads were washed sequentially with low-salt buffer, high-salt buffer, LiCl buffer and twice with TE and elution in 500 µl of elution buffer (1% SDS, 0.1 M NaHCO3). The elutes were heated at 42 °C for 2 h and treatment with proteinase K followed by 65 °C 10 h to reverse the cross-linking and were treated with RNaseA for 30 min before DNA was extracted and purified. The ChIP libraries were prepared using KAPA HyperPrep Kits (Roche, 07962347001), and then run on the Illumina sequencer Hiseq-Xten PE150.

**ChIP-seq analysis**. The Primary Analysis of ChIP-Seq Data sets were performed by using Illumina's Genome Analysis pipeline. The sequencing reads were aligned to the rat genome UCSC build rn6 by using Bowtie2 alignment programs in two ways: only uniquely aligned reads were kept or both uniquely aligned reads and the sequencing reads that align to repetitive regions were kept for downstream analysis (if a read aligns to multiple genome locations, only one location is arbitrarily chosen). The multiple reads were collapsed in order to reduce the PCR biases. The aligned reads were used for peak finding with HOMER[60,61] (http://biowhat.ucsd.edu/homer). The identification of ChIP-seq peaks was performed using HOMER following protocols as described previously[61]. For transcription factors, peaks were identified by searching locations of high read density using a 200-bp sliding window. Regions of maximal density exceeding a given threshold were called as peaks, and we required adjacent peaks to be at least 500 bp away to avoid redundant detection. Only one tag from each unique position was considered to avoid clonal artifacts from sequencing. The threshold for the number of tags that determined a valid peak was selected at a false discovery rate of 0.001 determined by peak finding using randomized tag positions in a genome with an effective size of $2 \times 10^9$ bp. We also required peaks to have at least 4-fold more tags (normalized to total count) than input control samples. In addition, we required 4-fold more tags relative to the local background region (10 kb) to avoid identifying regions with genomic duplications or nonlocalized binding. Annotated positions for promoters, exons, introns and other features were based on RefSeq transcripts and repeat annotations from the University of California, Santa Cruz. Peaks from separate experiments were considered equivalent/co-bound if their peak centers were located within 200 bp of each other. Read density heat maps were created by first using HOMER to generate read densities and then visualized using Java TreeView (http://jtreeview.sourceforge.net). To define the overlapping of two ChIP-seq, for instance of OLIG2 and H3K9me3, we first found out OLIG2 binding peak centers, then checked out H3K9me3 enrichment around these peak centers by using ComputeMatrix. And then visualized using PlotHeatmap. Motif analysis was carried out with HOMER.

**Quantification and statistical analysis**. All data presented are expressed as arithmetic mean ± SEM. All statistical analyses were performed using GraphPad Prism 7. Null hypotheses were rejected at p values equal to or higher than 0.05. Statistical significance was determined using two-tailed unpaired Student's t tests or one-way ANOVA. Quantifications were performed from at least three experimental groups in a blinded fashion. No statistical methods were used to pre-determine sample sizes, but our sample sizes are similar to those generally employed in the field. No randomization was used to collect all the data; they were quantified blindly.

**Reporting summary**. Further information on research design is available in the Nature Research Reporting Summary linked to this article.

## Data availability

The raw sequencing data generated in this study have been deposited in the National Center for Biotechnology Information (NCBI) BioProject database under BioProject: PRJNA673758. Source data are provided with this paper.

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

## Acknowledgements

We thank Dr. Jianqin Niu for technical supports. We thank William D. Richardson (University College London), Huiliang Li (University College London) and Lin Xiao (South China Normal University) for experimental resource. This study was funded by grants from the National Key R&D Program of China (2021YFA1101401 to W.M); and the Fundamental Research Funds for the Central Universities 20720190086 to W.M and 20720190072 to L.Z.; and National Natural Science Foundation of China (31872642 and 32170966 to L.Z.); and Natural Science Foundation of Fujian Province of China 2019J01573 to X.N.

## Author contributions

K.Z., S.C. Q.Y., S.G., Q.C., Z.L., L.L., H.L. and J.H. designed experiments and collected and analyzed data. K.Z., S.C. Q.Y. and S.G. performed primary OPCs culture experiments. Q.C., H.L. and J.H. performed mouse model associated experiments. K.Z., Z.L. and L.L. collected ChIP-seq and RNA-seq data. M.J., X.P., W.D., and N.X. analyzed ChIP-seq and RNA-seq data. K.Z. and Z.W. wrote the manuscript. B.W., Z.W., L.Z. and W.M. commented and edited the manuscript. N.X., L.Z. and W.M. provided funding. All of the authors discussed the results and commented on the manuscript.

## Competing interests

The authors declare no competing interests.
