## [Peer Review File · Nature Communications]

Reviewers' Comments:

Reviewer #1:

Remarks to the Author:

In this manuscript, Zhang et al. report that OLIG2 recruits the H3K9me3 methyltransferase SETDB1 to the Sox11 gene at the oligodendrocyte immature stage to silence Sox11 and thereby allow differentiation into mature oligodendrocytes. Although this study is interesting, many improvements need to be made, in the main text, Figures, figure legends, Methods, in the organization of the manuscript. In addition, important experiments/conditions are missing and there are a couple of overstatements. Here are below specific comments:

1/ In general, the use of English is very poor and needs to be improved, preferentially by a native English speaker.

2/ Introduction, first paragraph: Multiple sclerosis is not a mental disease, it is a neurodegenerative disease that can affect the entire central nervous system. Please, correct. Leukodystrophies are not mental diseases either, they are either developmental hypomyelinating diseases or demyelinating diseases with onset in adults. Some of them can lead to mental retardation, but not all of them, and the entire nervous system can be affected, depending on the gene mutation causing the disease.

3/ Introduction, first paragraph, last sentence : Please, change « remains unknown » to « remain partially known ».

4/ The work of the Wegner lab on Olig2 in oligodendrocyte differentiation and myelination is not cited, which is not acceptable.

5/ Figure 1h: The field of view of the image GFP/CC1 of the Olig2-VP64 does not perfectly correspond to the field of view of the individual images GFP and CC1, it is shifted to one side. Please, correct.

6/ Supplementary Fig. 1: some letters to describe each panel in the figure legend do not correspond to the figure. Please, adjust.

7/ The authors need to explain how they came up with the design of the dominant-activator form of Olig2. What is VP16 ? There is no reference to previous work (until the Discussion section) and no explanation. References to previous work and a short description need to be given already in the Results section.

8/ Page 6, last sentence : The data presented so far do not indicate, but only suggest, that OLIG2 recruits SETDB1 to govern H3K9me3 distribution for gene repression in early stage. Please, provide the evidence: show that H3K9me3 marks corresponding to OLIG2 binding sites are lost in cells where SETDB1 is ablated. Similar comments at the start of the following section: without providing the evidence above, the authors cannot assume that « the disintegration of repressive complex by SETDB1 inhibition is equivalent to depriving the role of OLIG2 in transcriptional repression ». In addition, even if the authors show that H3K9me3 marks corresponding to OLIG2 binding sites are lost in cells where SETDB1 is ablated, the ablation of SETDB1 is very likely to affect other regions that are not bound by OLIG2, unless the authors show that only OLIG2 binding sites loose H3K9me3 marks when SETDB1 is ablated, which is highly unlikely. Therefore, without further evidence, the ablation of SETDB1 cannot be interpreted as similar as OLIG2-mediated transcriptional repression. A possible way to go forward would be to completely re-organize the manuscript, focusing on the function of SETDB1 in myelination and remyelination and show that the function of SETDB1 is mediated at least partly by interacting with OLIG2, which recruits SETDB1 to a specific set of target genes.

Similar comment for conclusion page 9, lines 181-183: the myelination and remyelination defects of the Setdb1 mutant mice cannot be directly attributed to the repressive function of OLIG2.

9/ The authors used immature oligodendrocytes (iOLs) for many of their experiments, but it is not explained how they generate iOLs. In the Methods section, the culture medium used to induce OPC

differentiation is described, but no indication of timing is given to generate iOLs and mature OLs.

10/ Fig. 4h, low magnification: we do not see much. Please, increase the brightness of the images.

11/ In the remyelination experiments shown in Fig. 4, the onset of protein loss after tamoxifen injection is not described and does not seem to have been analyzed. It is essential to show when SETDB1 is lost after tamoxifen injection. Ideally, SETDB1 is lost before carrying out the lysolecithin lesion or at least before remyelination starts.

12/ Fig. 5c-d: the labeling is wrong.

13/ Page 9, lines 194-195: The conclusion here is again an overstatement. The findings that the expression of Enpp6 was declined in vitro and in vivo either by SETDB1 inhibition or OLIG2-VP64 overexpression does not indicate that « OLIG2-SETDB1 repressive complex is required for iOLs formation from OPCs », these are correlative observations that only suggest that this complex could be required for iOLs formation from OPCs.

14/ Page 11, lines 234-235: «the direct regulation of SOX11 by OLIG2-SETDB1 complex was verified by luciferase assay in 293T cells (Fig. 6h). » Luciferase gene reporter assays do not prove a direct regulation, they just show a regulation, direct or indirect. However, the combination of binding to the Sox11 promoter by ChIP together with the regulation of activity of the Sox11 promoter by luciferase gene reporter assay give a strong indication of direct regulation. So, this needs to be re-formulated.

In addition :

- in Fig. 6g, SETDB1 occupancy on the Sox11 gene is missing. Without this, no conclusion can be made.
- In Fig. 6h, SETDB1 overexpression alone is missing. Without this condition, no conclusion can be made.
- The luciferase gene reporter assay should also be made on the Sox11 promoter where OLIG2 binding site is deleted.

15/ The figure legends are not always detailed enough, so it is not always straight forward to understand what is shown. For example, in Fig. 6h, please explain what SETDB1 M is. By going through the main text, we understand that SETDB1 M is a mutant without enzymatic activity, but this should be already clear in the figure legend. In addition, there is no description of the constructs in the Methods section, except for the luciferase constructs, which there again are not described enough (which regulatory region is cloned ? references are missing). The sequences of the Sox11 ASO used are also missing.

16/ How is the survival of P3 pups after tamoxifen injection? How can the authors carry out i.p. injections at P3? Usually, at this age, subcutaneous injections are used because there is no space to inject anything i.p. Another possibility is to give tamoxifen through mother's milk.

Reviewer #2:

Remarks to the Author:

The manuscript by Zhang et al. suggested the oligodendrocyte lineage master regulator, OLIG2, plays a dual role in both activating genes for myelination and suppressing genes that prevent lineage progression. While the activating role of OLIG2 has been demonstrated previously by Yu et al., 2013, the authors aimed to provide the complementary picture by suggesting the repressive role of OLIG2 in recruiting H3K9 histone methyltransferase (HMT), Setdb1, to genomic loci of genes that are lineage inappropriate. Furthermore, the authors used a variety of in vitro and in vivo approaches to selectively ablating Setdb1 along OL lineage and demonstrated the necessity of Setdb1 in development and repair for proper myelination. While the overall concept of the manuscript brings new insight of the role of OLIG2 in master regulating the epigenetic landscape during oligodendrocyte development, the main point of the conclusion is only correlative, and not supported by sufficient experimental evidence.

1. The critical role of H3K9me3 for OL differentiation and its genomic distribution has been extensively characterized by previous publication (Liu et al. J Neurosci 2015) and further validated in the manuscript. Therefore, it is not surprising that deletion of Setdb1, the writer for this modification, led to critical consequences in OPC differentiation and subsequent myelination. However, HMTs exert high redundant functional roles. It is important to note that IP of OLIG2 revealed several H3K9 HMTs were in the same complex (Fig.2a) and some of them even bear higher statistical significance than Setdb1. The authors decided to focus on the interaction of OLIG2 and Setdb1 exclusively based on the lack of transcriptional consequence in silencing other HMTs (fig 2c). However, these results are in contradictory to Liu et al, 2015, which has demonstrated the transcriptional, translational and functional consequences of silencing other H3K9 HMTs. While results from both groups may not be mutually exclusive, IP of OLIG2 with other H3K9 HMTS should be performed. Furthermore, in all of the transgenic mice, it is essential to show if H3K9me3 is affected, given the redundant roles of other K9 HMTs. If not, the effect of ablating Setdb1 may be independent from its role in regulating the epigenome.

2. The assembly of OLIG2/Setdb1 was suggested by bioinformatics analysis of overlapping of Olig2 ChIP-seq and H3K9me3 ChIP-seq. However, there are some major concerns in this analysis. First of all, the overlapping of OLIG2 and H3K9me3 ChIP-seq is not sufficient to demonstrate the assembly of OLIG2 and Setdb1, as H3K9me3 could be mediated by many other H3K9 HMTs, which are also present in the complex containing OLIG2 (as mentioned above). Therefore, ChIP-seq of Setdb1 and overlap with OLIG2 ChIP-seq is essential to make the connection. Second, HOMER identified OLIG2 binding motif in H3K9me3 enriched regions in iOL and iOL-specific cells, but not in OPC. Is OLIG2 the only binding motif identified? Again, almost same type of analysis from Liu et al., 2015 reported several other binding motifs/transcription factor (but not OLIG2) in iOL and their association with the repressive complex. How can the authors exclude the possibility that H3K9me3 was not added by other HMTs in the complex by other transcription factors? Or Setdb1 was recruited by these other transcription factors reported from Liu et al.? Third, very little technical information was provided for how the bioinformatics analysis is performed. Is it performed with published datasets or generated by the authors themselves? How is overlapping defined? This is very concerning, given that Fig. 2l and 2m showed similar enrichment of H3K9me3 (statistical analysis?) at Olig2 binding site in both OPC and iOL. It is intriguing to think why Olig2 motif was not detected in OPC (Fig.2j).

3. While the authors used a variety of transgenic mouse lines to demonstrate the necessity of Setdb1 in oligodendrocyte development, these results are largely descriptive and lack mechanistic insight. The authors attribute the role of Setdb1 to silent inhibitors of differentiation by adding H3K9me3 to those loci (fig.6), which is a well characterized role of repressive histone modifications by many research groups. Therefore these findings are not surprising and did not provide a major advancement of knowledge in the community. In addition, the ontology analysis from OLIG2 regulated genes and H3K9me3 enriched genes (fig.2) revealed several other groups of genes bearing higher statistical significance (e.g. regulation of neuron differentiation). What are the levels of those genes in the Setdb1 cKO mice? It is definitely worth checking.

4. The authors described the use of Setdb1 conditional knockout mice as to selectively depriving the repressive role of OLIG2. This is an intriguing point. If this is the case, one would expect that, the master regulator of OLIG2 would exert more activating roles in these transgenic mice, thereby, perhaps, leading to over activation of myelinogenesis genes and the opposite phenotype as they observed.

5. Experiments related to Olig2-VP64 constructs are confusing and difficult to interpret. While the normal levels of OLIG2 have no effects on differentiation, over activation of OLIG2 induced reduced OPC differentiation. What exactly does VP64 domain do? It induced enhanced binding of OLIG2 to DNA? If so, why would it only override the inhibitory function, but not activating function of OLIG2? Some proper controls are also missing, such as a VP64 domain alone construct, or VP64 domain fused with another transcription factor.

6. The authors should rephrase when they describe H3K9me3 enrichment, and not use "H3K9me3 binding", as this is a histone modification and enriched over a large region, not a transcription factor, which binds to DNA sequences.

7. The authors should check the statistical analysis method used. Only t-test was used throughout the manuscript, while clearly there were cases ANOVA should be applied (e.g Fig.6, Supp Fig.7)

8. Supp fig1. legend was mislabeled.

Overall, the authors provided ample evidence of function of Setdb1 in oligodendrocyte lineage in vivo, but its relationship with OLIG2's repressive role remains unclear.

Reviewer #3:

Remarks to the Author:

This manuscript by Zhang et al investigated the repressive function of Olig2 in myelination and found that SETDB1 was recruited for distributing H3K9me3 deposition. Further the authors identified SETDB1 is essential for the repressive role of Olig2 and deletion of SETDB1 inhibits oligodendrocyte myelination and remyelination, and SETDB1 executes its function probably by directly regulating Sox11 expression in oligodendroglia. It has been well documented that Olig2 is an important transcriptional factor for oligodendroglial differentiation and myelination that can directly initial a number of myelin related genes. This manuscript described a yet unknown function of Olig2 in repressing negative regulators. The findings are novel and provide an important advance in understanding the role of Olig2 and SETDB1 complex.

There are several concerns that should be addressed.

1. The authors proposed the SETDB1 and Olig2 complex functions "in early stage but not in later stage nor the maintenance of myelin...", regarding that the SETDB1 deletion driven by different oligodendroglia-specific Cre(Ert) lines displayed variable phenotypes of myelination (i.g, Figure3, Figure 5). But this claim is vacant without examining the expression pattern of SETDB1 in oligodendroglial lineage cells during development and adulthood. For example, whether the SETDB1 expression is stage-dependent? Or age-dependent? In addition, the deletion of SETDB1 in OPCs and OLs needs to be assessed in the SETDB1 cKO lines, which may provide clues in understanding the inconsistency of the myelination phenotypes.

2. The results showed that both mature and immature OL densities were greatly decreased in the SETDB1 cKO mice driven by Olig1-cre, NG2-cre or PDGFRa-CreErt (Fig. 3-5). The data of oligodendroglia apoptosis upon SETDB1 deletion needs to be shown and included, so that to prove the role of SETDB1 in regulating oligodendroglial differentiation and myelination.

3. Sox11 was identified as the target genes of SETDB1 complex, and it is important to show whether Sox11 deletion in OPCs can attenuate the effect of SETDB1 deletion and so that rescue the myelination deficits. Thus, quantifying the MBP expression and mature OL numbers in the Sox11 cKO mice is essential for this claim (Figure 6i, supplementary Figure 7i).

4. As it seems myelination was enhanced in the brains with Sox11 knockdown (supplementary Figure 7i), the infective efficiency needs to be examined.

5. In supplementary Figure 7e, quantification of Hes1 knockdown and CNP expression is required to prove the Hes1 function in altering OPC differentiation.

6. How does the Olig2 -VP64 (transcriptional activator) solely display enhanced repressive functions? As Olig2 activates pro-myelination genes. This needs further clarification and comments.

7. Clearly state "N" number the figure legend.

Responds to the reviewer's comments:

Reviewer #1 (Remarks to the Author):

In this manuscript, Zhang et al. report that OLIG2 recruits the H3K9me3 methyltransferase SETDB1 to the Sox11 gene at the oligodendrocyte immature stage to silence Sox11 and thereby allow differentiation into mature oligodendrocytes. Although this study is interesting, many improvements need to be made, in the main text, Figures, figure legends, Methods, in the organization of the manuscript. In addition, important experiments/conditions are missing and there are a couple of overstatements. Here are below specific comments:

1/ In general, the use of English is very poor and needs to be improved, preferentially by a native English speaker.

Reply: Thanks for the reviewer's suggestion. We revised the manuscript and corrected some grammar mistakes and inappropriate formulations.

2/ Introduction, first paragraph: Multiple sclerosis is not a mental disease, it is a neurodegenerative disease that can affect the entire central nervous system. Please, correct. Leukodystrophies are not mental diseases either, they are either developmental hypomyelinating diseases or demyelinating diseases with onset in adults. Some of them can lead to mental retardation, but not all of them, and the entire nervous system can be affected, depending on the gene mutation causing the disease.

Reply: The reviewer is right and we replaced the statement "mental disease" with "neurological disease".

3/ Introduction, first paragraph, last sentence: Please, change « remains unknown » to « remain partially known ».

Reply: We amended the sentence according to the reviewer's suggestion.

4/ The work of the Wegner lab on Olig2 in oligodendrocyte differentiation and myelination is not cited, which is not acceptable.

Reply: Thank the reviewer for this reference. Kuspert et al (2015) did a nice work on elaborating the mechanism of OLIG2 regulating myelinogenesis.

5/ Figure 1h: The field of view of the image GFP/CC1 of the Olig2-VP64 does not perfectly correspond to the field of view of the individual images GFP and CC1, it is shifted to one side. Please, correct.

Reply: We corrected the composition according to the reviewer's suggestion.

6/ Supplementary Fig. 1: some letters to describe each panel in the figure legend do not correspond to the figure. Please, adjust.

Reply: It's our mistake. Thanks for the reviewer's correction. And we corrected the figure legend.

7/ The authors need to explain how they came up with the design of the dominant-activator form of Olig2. What is VP16 ? There is no reference to previous work (until the Discussion section) and no explanation. References to previous work and a short description need to be given already in the Results section.

Reply: VP64 is 4 tandem of VP16, a herpes simplex virus protein that could function as transactivation domain (Novitch et al., 2001, Marshall et al., 2005). OLIG2-VP64 can abolish the transcriptional inhibition function by transforming OLIG2 into transcriptional activation form. Thanks for the reviewer's suggestion. We added this description and reference in the manuscript.

8/ Page 6, last sentence : *The data presented so far do not indicate, but only suggest, that OLIG2 recruits SETDB1 to govern H3K9me3 distribution for gene repression in early stage. Please, provide the evidence: show that H3K9me3 marks corresponding to OLIG2 binding sites are lost in cells where SETDB1 is ablated. Similar comments at the start of the following section: without providing the evidence above, the authors cannot assume that « the disintegration of repressive complex by SETDB1 inhibition is equivalent to depriving the role of OLIG2 in transcriptional repression ». In addition, even if the authors show that H3K9me3 marks corresponding to OLIG2 binding sites are lost in cells where SETDB1 is ablated, the ablation of SETDB1 is very likely to affect other regions that are not bound by OLIG2, unless the authors show that only OLIG2 binding sites loose H3K9me3 marks when SETDB1 is ablated, which is highly unlikely. Therefore, without further evidence, the ablation of SETDB1 cannot be interpreted as similar as OLIG2-mediated transcriptional repression. A possible way to go forward would be to completely re-organize the manuscript, focusing on the function of SETDB1 in myelination and remyelination and show that the function of SETDB1 is mediated at least partly by interacting with OLIG2, which recruits SETDB1 to a specific set of target genes.*

Similar comment for conclusion page 9, lines 181-183: the myelination and remyelination defects of the *Setdb1* mutant mice cannot be directly attributed to the repressive function of OLIG2.

Reply: It's a good suggestion and we performed H3K9me3 ChIP-seq on iOLs from wild type or *Setdb1* mutant mouse brain. As the result shown below (please see Supplementary Fig. 7a), H3K9me3 peak density was almost diminished around gene loci occupied by OLIG2. So we think that OLIG2 recruit SETDB1 to establish H3K9me3 modification on some gene loci that need to be repressed.

For some loose statements, we corrected according to the reviewer's suggestion.

9/ The authors used immature oligodendrocytes (iOLs) for many of their experiments, but it is not explained how they generate iOLs. In the Methods section, the culture medium used to induce OPC differentiation is described, but no indication of timing is given to generate iOLs and mature OLs.

Reply: For iOLs, we harvest cells at 24h after T3 induction. And for mOLs, we induce OPCs differentiation for 72h. And we added these descriptions in the methods part.

10/ Fig. 4h, low magnification: we do not see much. Please, increase the brightness of the images.

Reply: Thanks for the reviewer's question. We adjusted the brightness of the images in fig. 4h.

11/ In the remyelination experiments shown in Fig. 4, the onset of protein loss after tamoxifen injection is not described and does not seem to have been analyzed. It is essential to show when SETDB1 is lost after tamoxifen injection. Ideally, SETDB1 is lost before carrying out the lysolecithin lesion or at least before remyelination starts.

Reply: To examine the efficiency of tamoxifen induction, *Sox10-CreER; Setdb1^{F/F}* Transgenic mice crossed with *rosa26-tomato* mice for labelling OPC lineage cells.

After 3-days tamoxifen induction, SETDB1 was undetectable in OPC lineage cells , indicating that our tamoxifen induction system is high efficiency in *Setdb1* knocking out (please see Supplementary Fig. 6p).

12/ Fig. 5c-d: the labeling is wrong.

Reply: Thank the review to point it out. Thanks for the reviewer's correction. And we corrected the labeling.

13/ Page 9, lines 194-195: The conclusion here is again an overstatement. The findings that the expression of Enpp6 was declined in vitro and in vivo either by SETDB1 inhibition or OLIG2-VP64 overexpression does not indicate that « OLIG2-SETDB1 repressive complex is required for iOLs formation from OPCs », these are correlative observations that only suggest that this complex could be required for iOLs formation from OPCs.

Reply: The reviewer is right. We have corrected the sentences as follows: “Thus, OLIG2-SETDB1 repressive complex could be required for iOLs formation from OPCs.”

14/ Page 11, lines 234-235: «the direct regulation of SOX11 by OLIG2-SETDB1 complex was verified by luciferase assay in 293T cells (Fig. 6h). » Luciferase gene reporter assays do not prove a direct regulation, they just show a regulation, direct or indirect. However, the combination of binding to the Sox11 promoter by ChIP together with the regulation of activity of the Sox11 promoter by luciferase gene reporter assay give a strong indication of direct regulation. So, this needs to be re-formulated.

In addition :

- in Fig. 6g, SETDB1 occupancy on the Sox11 gene is missing. Without this, no conclusion can be made.
- In Fig. 6h, SETDB1 overexpression alone is missing. Without this condition, no conclusion can be made.
- The luciferase gene reporter assay should also be made on the Sox11 promoter where OLIG2 binding site is deleted.

Reply: Thank the reviewer for the question. We added the ChIP experiments to further prove the direct regulation of Sox11 by Olig2-Setdb1 complex and reworded this part.

(1) We changed it to “the regulation of SOX11 by OLIG2-SETDB1 complex was verified by luciferase assay in 293T cells.”

(2) We checked out H3K9me3 enrichment on Sox11 promoter region. In Setdb1 mutant iOLs, H3K9me3 enrichment is dramatically reduced on Sox11 promoter region (please see Supplementary Fig. 7g).

(3) To clarify SETDB1 occupancy on the Sox11 promoter region, we performed HA-SETDB1 ChIP-qPCR assay. When we knocked down Olig2 in iOLs, there is less SETDB1 binding to promoter region of Sox11 (please see Supplementary Fig. 7h).

(4) We added the condition “SETDB1 overexpression” alone in Fig.6i.

(5) According to our result in Fig.6h, OLIG2 binds on the whole promoter region of *Sox11* (-1kb – 0). So, instead of deleting the *Sox11* promoter, we chose to mutate two conservative OLIG2 binding motif. Then we performed luciferase assay. Ablation of Olig2 binding motif impair the effect of Olig2 and Setdb1 on Sox11-driven luciferase expression (please see Supplementary Fig. 7j).

15/ The figure legends are not always detailed enough, so it is not always straight forward to understand what is shown. For example, in Fig. 6h, please

explain what SETDB1 M is. By going through the main text, we understand that SETDB1 M is a mutant without enzymatic activity, but this should be already clear in the figure legend. In addition, there is no description of the constructs in the Methods section, except for the luciferase constructs, which there again are not described enough (which regulatory region is cloned ? references are missing). The sequences of the Sox11 ASO used are also missing.

Reply: It's a good suggestion and we added some essential information in main text and figure legend of Fig. 6i.

In Fig 6i, SETDB1 Cys798 is a vital amino acid to its methyltransferase activity. And we added the information of ASO in the methods. Second generation ASO was designed to target mouse Sox11. ASO consisted of 18-mer nucleotide sequences (5'-ttattactctattggcc-3') with 2'-O-methoxyethyl (MOE) modifications and a mixed backbone of phosphorothioate and phosphodiester internucleotide linkages. We also described how we constructed the Sox11-luciferase assay in methods.

16/ How is the survival of P3 pups after tamoxifen injection? How can the authors carry out i.p. injections at P3? Usually, at this age, subcutaneous injections are used because there is no space to inject anything i.p. Another possibility is to give tamoxifen through mother's milk.

Reply: It's our mistake in writing. To administrate tamoxifen to P3 pups, we carried out subcutaneous injection on the pups' back. We added this information in the methods.

Reviewer #2 (Remarks to the Author):

The manuscript by Zhang et al. suggested the oligodendrocyte lineage master regulator, OLIG2, plays a dual role in both activating genes for myelination and suppressing genes that prevent lineage progression. While the activating role of OLIG2 has been demonstrated previously by Yu et al., 2013, the authors aimed to provide the complementary picture by suggesting the repressive role of OLIG2 in recruiting H3K9 histone methyltransferase (HMT), Setdb1, to genomic loci of genes that are lineage inappropriate. Furthermore, the authors used a variety of in vitro and in vivo approaches to selectively ablating Setdb1 along OL lineage and demonstrated the necessity of Setdb1 in development and repair for proper myelination. While the overall concept of the manuscript brings new insight of the role of OLIG2 in master regulating the epigenetic landscape during oligodendrocyte development, the main point of the conclusion is only correlative, and not supported by sufficient experimental evidence.

1. The critical role of H3K9me3 for OL differentiation and its genomic distribution has been extensively characterized by previous publication (Liu et al. J Neurosci 2015) and further validated in the manuscript. Therefore, it is not surprising that deletion of Setdb1, the writer for this modification, led to critical

consequences in OPC differentiation and subsequent myelination. However, HMTs exert high redundant functional roles. It is important to note that IP of OLIG2 revealed several H3K9 HMTs were in the same complex (Fig.2a) and some of them even bear higher statistical significance than Setdb1. The authors decided to focus on the interaction of OLIG2 and Setdb1 exclusively based on the lack of transcriptional consequence in silencing other HMTs (fig 2c). However, these results are in contradictory to Liu et al, 2015, which has demonstrated the transcriptional, translational and functional consequences of silencing other H3K9 HMTs. While results from both groups may not be mutually exclusive, *IP of OLIG2 with other H3K9 HMTs should be performed*. Furthermore, in all of the transgenic mice, it is essential to show if H3K9me3 is affected, given the redundant roles of other K9 HMTs. If not, the effect of ablating Setdb1 may be independent from its role in regulating the epigenome.

Reply: Thank the reviewer for the question.

(1) Liu et al. performed in vitro experiments in mouse OPCs and iOLs, while we used rat OPCs and iOLs to carry out experiments. These may cause some difference in our results. Furthermore, Liu et al. examined role of several HMTs in OPCs differentiation, however they omitted SETDB1 when they performed assays. And we found that SETDB1 is the most dominant HMT in OPC development exactly, as shown in Fig. 2c.

(2) HMTs contains 5 members, which is EHMT1, EHMT2, SUV39H1, SUV39H2, SETDB1. EHMT2 is specifically associated with H3K9me and H3K9me2 and don't catalyze H3K9me3. And SUV39H2 even don't express in OPC lineage cells, which is reported by Barres lab (Zhang et al. J Neurosci 2014). So besides for SETDB1, we checked interaction of OLIG2 and EHMT1 or SUV39H1. As the result shown below, compared to the interaction between Olig2 and Setdb1, there is very weak interaction between OLIG2 and EHMT1 or SUV39H1 (please see Supplementary Fig. 2c-2f).

(3) To verify the key role of SETDB1 to H3K9me3, we assessed H3K9me3 level in *Setdb1* mutant brain with immunostaining. The result showed that H3K9me3 level is obviously reduced in in mutant OPC lineage cells, labeled with *rosa26-tomato*, suggesting that H3K9me3 is influenced by SETDB1 to a great extent in OPC lineage cells (please see Supplementary Fig. 3b).

2. The assembly of OLIG2/Setdb1 was suggested by bioinformatics analysis of overlapping of Olig2 ChIP-seq and H3K9me3 ChIP-seq. However, there are some major concerns in this analysis. First of all, the overlapping of OLIG2 and H3K9me3 ChIP-seq is not sufficient to demonstrate the assembly of OLIG2 and Setdb1, as H3K9me3 could be mediated by many other H3K9 HMTs, which are also present in the complex containing OLIG2 (as mentioned above). Therefore, *ChIP-seq of Setdb1 and overlap with OLIG2 ChIP-seq is essential to make the connection*. Second, HOMER identified OLIG2 binding motif in H3K9me3 enriched regions in iOL and iOL-specific cells, but not in OPC. Is OLIG2 the only binding motif identified? Again, almost same type of analysis from Liu et al., 2015 reported several other binding motifs/transcription factor (but not OLIG2) in iOL and their association with the repressive complex. How can the authors exclude the possibility that H3K9me3 was not added by other HMTs in the complex by other transcription factors? Or Setdb1 was recruited by these other transcription factors reported from Liu et al.? Third, very little technical information was provided for how the bioinformatics analysis is performed. Is it performed with published datasets or generated by the authors themselves? How is overlapping defined? This is very concerning, given that Fig. 2I and 2m showed similar enrichment of H3K9me3 (statistical analysis?) at Olig2 binding site in both OPC and iOL. It is intriguing to think why Olig2 motif was not detected in OPC (Fig.2j).

Reply: Thank the reviewer for the question.

(1) We tried to perform SETDB1 ChIP-seq in iOLs but failed. Perhaps because SETDB1 protein is vulnerable to formaldehyde fixation, and can't be recognized by antibodies. So we transduced HA-SETDB1 into rat iOLs, and performed HA-probe ChIP-seq instead. As the result shown below, SETDB1 recruited by OLIG2 on the genome of iOLs (please see Supplementary Fig. 2j).

(2) The reviewer is right. Indeed, OLIG2 was not the unique transcriptional factor, which was enriched by Motif analysis of H3K9me3 ChIP-seq. SOX10 was also found in the result. However, we found that OLIG2's binding motif was apparently increased after OPCs differentiation. Other transcriptional factor in OPC lineage cells, like SOX10, YY1, were not. Then we further examine the interaction between SETDB1 and a series of transcriptional factors enriched in oligodendrocytes. The result showed that there were weak or none interaction between them (please see Supplementary Fig. 2k).

(3) Except the sequencing data in the beginning of the main text, ChIP-seq data was product by our lab and sequencing corporation. The details of data analysis were added in methods.

(4) The conservational binding motif of OLIG2 exist in both H3K9me3 enrichment region of OPCs and iOLs. But in OPCs, abundance of OLIG2 motif is almost the same in H3K9me3 enrichment region and whole genome. So there is no statistic difference in OPCs.

3. While the authors used a variety of transgenic mouse lines to demonstrate

the necessity of Setdb1 in oligodendrocyte development, these results are largely descriptive and lack mechanistic insight. The authors attribute the role of Setdb1 to silent inhibitors of differentiation by adding H3K9me3 to those loci (fig.6), which is a well characterized role of repressive histone modifications by many research groups. Therefore these findings are not surprising and did not *provide a major advancement of knowledge in the community*. In addition, the ontology analysis from OLIG2 regulated genes and H3K9me3 enriched genes (fig.2) revealed several other groups of genes bearing higher statistical significance (e.g. regulation of neuron differentiation). What are the levels of those genes in the Setdb1 cKO mice? It is definitely worth checking.

Reply: Our study focused on repressive function of Olig2 in myelination. The previous research on OLIG2 in OPC is mostly limited to its transcriptional activator function. We described a dual role of OLIG2 in OPC development, which is novel and provides a major advancement of knowledge in OL field.

We checked GO result of RNA-seq in Setdb1 cKO iOLs, and found the “regulation of neuron differentiation” term. But the p-value is not low enough, so it’s not shown in the figure.

4. The authors described the use of Setdb1 conditional knockout mice as to selectively depriving the repressive role of OLIG2. This is an intriguing point. If this is the case, one would expect that, the master regulator of OLIG2 would exert more activating roles in these transgenic mice, thereby, perhaps, leading to over activation of myelinogenesis genes and the opposite phenotype as they observed.

Reply: we consider that master regulator OLIG2 recruits different cofactors on different loci. In this condition, OLIG2 promotes some gene expression, such as *Sox10*, and represses some gene in another region, such as *Sox11*. These happened in the same time but independent in space of genome. When SETDB1 is lacked, only these repressed gene sites can be affected. For instance *Sox11* is activated in *Setdb1* cKO, and then inhibits the whole differentiation process.

5. Experiments related to Olig2-VP64 constructs are confusing and difficult to interpret. While the normal levels of OLIG2 have no effects on differentiation, over activation of OLIG2 induced reduced OPC differentiation. What exactly does VP64 domain do? It induced enhanced binding of OLIG2 to DNA? If so, why would it only override the inhibitory function, but not activating function of OLIG2? Some proper controls are also missing, such as a VP64 domain alone construct, or VP64 domain fused with another transcription factor.

Reply: VP64 is 4 tandem of VP16, a herpes simplex virus protein that could function as transactivation domain (Novitsch et al., 2001, Marshall et al., 2005). OLIG2-VP64 can abolish the transcriptional inhibition function by transforming OLIG2 into transcriptional activation form. Thanks for the reviewer’s suggestion. We added this description and reference in the manuscript.

We also checked the effect of VP64 alone to OPC differentiation. As the result shown

below, VP64 alone almost did not active the process of OPC differentiation (please see Supplementary Fig. 1d).

6. The authors should rephrase when they describe H3K9me3 enrichment, and not use “H3K9me3 binding”, as this is a histone modification and enriched over a large region, not a transcription factor, which binds to DNA sequences.

Reply: The reviewer is right. We correct it as the reviewer suggested.

7. The authors should check the statistical analysis method used. Only t-test was used throughout the manuscript, while clearly there were cases ANOVA should be applied (e.g Fig.6, Supp Fig.7)

Reply: Thank the review’s advice. We checked the statistical analysis method used, and replaced some misused t-test with ANOVA.

8. Supp fig1. legend was mislabeled.

Reply: It’s our mistake. We adjusted the order according to the reviewer’s suggestion.

Overall, the authors provided ample evidence of function of Setdb1 in oligodendrocyte lineage in vivo, but *its relationship with OLIG2’s repressive role remains unclear*.

Reviewer #3 (Remarks to the Author):

This manuscript by Zhang et al investigated the repressive function of Olig2 in myelination and found that SETDB1 was recruited for distributing H3K9me3 deposition. Further the authors identified SETDB1 is essential for the repressive role of Olig2 and deletion of STEDB1 inhibits oligodendrocyte myelination and remyelination, and STEDB1 executes its function probably by directly regulating

Sox11 expression in oligodendroglia. It has been well documented that Olig2 is an important transcriptional factor for oligodendroglial differentiation and myelination that can directly initial a number of myelin related genes. This manuscript described a yet unknown function of Olig2 in repressing negative regulators. The findings are novel and provide an important advance in understanding the role of Olig2 and SETDB1 complex.

There are several concerns that should be addressed.

1. The authors proposed the SETDB1 and Olig2 complex functions “in early stage but not in later stage nor the maintenance of myelin...”, regarding that the SETDB1 deletion driven by different oligodendroglia-specific Cre(Ert) lines displayed variable phenotypes of myelination (i.g, Figure3, Figure 5). But this claim is vacant without examining the expression pattern of SETDB1 in oligodendroglial lineage cells during development and adulthood. For example, whether the SETDB1 expression is stage-dependent? Or age-dependent? In addition, the deletion of SETDB1 in OPCs and OLs needs to be assessed in the SETDB1 cKO lines, which may provide clues in understanding the inconsistency of the myelination phenotypes.

Reply: Thank the review’s advice.

(1) To explore whether the SETDB1 expression is age-dependent, we collected brain samples of different age mice. As the immunostaining shown, the protein level of SETDB1 is reduced when mice get older (please see Supplementary Fig. 2i).

(2) To examine the knock-out efficiency of *Setdb1* in tamoxifen inducible system, we carried out 3-days tamoxifen administration on *Sox10-CreER; Setdb1^{F/F}; rosa26-tomato* mice. Immunostaining result showed that *Setdb1* expression was essentially eliminated in the brain of induced conditional knockout mice (please see Supplementary Fig. 6p).

2. The results showed that both mature and immature OL densities were greatly decreased in the SETDB1 cKO mice driven by Olig1-cre, NG2-cre or PDGFRa-CreErt (Fig. 3-5). The data of oligodendroglia apoptosis upon SETDB1 deletion needs to be shown and included, so that to prove the role of SETDB1 in regulating oligodendroglial differentiation and myelination.

Reply: It's a good suggestion and we performed TUNEL staining after *Setdb1* was knocked out. As the result shown, there is no obvious dead cell in *Setdb1* deficient brain (please see Supplementary Fig. 6r).

3. Sox11 was identified as the target genes of SETDB1 complex, and it is

important to show whether Sox11 deletion in OPCs can attenuate the effect of STEDB1 deletion and so that rescue the myelination deficits. Thus, quantifying the MBP expression and mature OL numbers in the Sox11 cKO mice is essential for this claim (Figure 6i, supplementary Figure 7i).

Reply: We added the statistical diagrams as the reviewer suggested.

4. As it seems myelination was enhanced in the brains with Sox11 knockdown (supplementary Figure 7i), the infective efficiency needs to be examined.

Reply: This is a good suggestion. We examined the efficiency by injection an ASO with a Cy3 fluorescence tag in the 5' end into Lateral ventricles of mice. A Cy3-labeled ASO was used to examine efficiency. As the result shown, the efficiency of ASO is preferably high (please see Supplementary Fig. 7l).

5. In supplementary Figure 7e, quantification of Hes1 knockdown and CNP expression is required to prove the Hes1 function in altering OPC differentiation. Reply: According to the reviewer's suggestions, we explored the role of *Hes1* in altering OPC differentiation. As the qPCR result shown, myelin genes *Mbp* and *Mag* is highly activated upon *Hes1* knocked out (please see Supplementary Fig. 7d).

6. How does the Olig2 -VP64 (transcriptional activator) solely display enhanced repressive functions? As Olig2 activates pro-myelination genes. This needs further clarification and comments.

Reply: VP64 is 4 tandem of VP16, a herpes simplex virus protein that could function as transactivation domain (Novitch et al., 2001, Marshall et al., 2005). OLIG2-VP64 can abolish the transcriptional inhibition function by transforming OLIG2 into transcriptional activation form. Thanks for the reviewer's suggestion. We added this description and reference in the manuscript.

7. Clearly state "N" number the figure legend.

Reply: Thank the review's advice. We added "N" number in the figure legend when necessary.

Reviewers' Comments:

Reviewer #1:

Remarks to the Author:

The authors have made a great effort to convincingly address all my questions and comments and, to my opinion, also the ones of the other reviewers. The manuscript is now a lot improved and the conclusions of the authors strengthened.

By going through their answers to the other reviewers, I however need to comment on the answer (2) to comment 1. of Reviewer #2 : There are many more HMTs than the 5 listed by the authors in their answer, but I assume that they mean 5 H3K9 HMTs. To my knowledge, there is however at least one more H3K9 HMT called PRDM2, which can also trimethylate H3K9. To fully answer comment 1. of Reviewer #2, PRDM2 expression and co-IP with Olig2 would therefore also need be checked.

Reviewer #2:

Remarks to the Author:

The revised manuscript remains largely the same as the previous version with limited attempts to provide additional evidence for the major conclusion. The additional experiments as well as the rebuttal points were insufficient and could not address the major concerns raised previously. The repressive role of Olig2 in regards to epigenetic regulation remains elusive and confusing. Several key technical details are still missing. The manuscript also bears several careless oversight. I suggest the authors to carefully proof read the manuscript.

1. One key experiment missing in the previous manuscript is the ChIP-seq of SETDB1 and its overlap with Olig2 peaks. The author claimed some technical difficulties with Setdb1 ChIP-seq and therefore proceeded with an exogenously expressed molecule through HA-tag. While exogenously expressed protein may not faithfully representing the physiological function, this experiment is a plus nevertheless. However, several key elements are missing: a) same experiments should be performed in OPC, (b) peak alignment between Setdb1 peak and Olig2 peak should be evaluated in both OPC and iOL (c) motif analysis on Setdb1 peaks in both OPC and iOL. These experiments are essential. Furthermore, if Olig2 was inducing transcriptional repression by recruiting Setdb1 upon differentiation, one would expect the enrichment of H3K9me3 around Olig2 binding site is only present in iOL but not in OPC. This is NOT what is shown in Fig 2l and 2m. In addition, to exclude the possibility that Setdb1 was recruited by other transcription factor, the co-IP experiments should be performed in the same cell type during differentiation, because these interactions are expected to cell type specific and lineage specific.

2. One of the other major concerns is the work here presented different, in some cases, contradictory findings to the work from the Casaccia lab, who documented the role of H3K9me3 in OPC differentiation and myelination earlier (Liu et al. J Neurosci. 2015). A few examples include the effect of knocking down HMTs in OPC and the peak binding motif analysis. Given the two studies are highly relevant, the work from the Casaccia lab should be cited, and if possible, compared side by side. Discrepancies, if any, should be evaluated and discussed. The authors attributed the discrepancies between their study and Liu et al. to different species used. It is important to note that Liu et al. used both rat and mouse OPCs for their in vitro study, with ChIP-seq performed using the same species. Therefore, it is quite unlikely as both groups are examining key differentiation regulators and lineage genes. Other possibilities should also be evaluated. For example, the efficiency of HMT knockdown is ~50% by transcripts in the current study (SuppFig.2b). What about protein levels? Are other HMTs bearing longer half-life therefore rendering them to remain functional longer than SETDB1 after silencing? When was differentiation induced after silencing? The authors claimed to focus on SETDB1 because "SETDB1 is the most dominant HMT in OPC development exactly, as shown in Fig. 2c". This is not what is shown in Fig.2c. While the interaction between Olig2 and Setdb1 was performed under physiological conditions, the interaction between Olig2 and other HMTs were performed with exogenously expressed HMTs, and in a non-relevant cell line, 293T cells. Wouldn't one expect these interactions to be cell type-specific and lineage-specific? Finally, the level of H3K9me3 should be quantified in all Setdb1 cKO

mice. For example, in SuppFig 3b, there are clearly many TdTomato+ OL that have H3K9me3. The tools and methods used for motif analysis are not discussed in the method section either.

3. Using various transgenic mouse models, overexpressing constructs and demyelination model, the function of Setdb1 in OPC differentiation and myelination is convincing, except when the authors attempt to link to the repressive function of Olig2. For example, as a master regulator activating many of the lineage progression genes, the activating role of Olig2 is overwritten due to the loss of its repressive role by the lack of Setdb1 in cKO mice. This is a very intriguing point. Why? What about the interaction between Olig2 and chromatin remodeling enzymes as Yu et al. demonstrated previously (e.g. Brg1)? What happens to those loci regulated by Olig2's activating role? Similarly, the VP64 constructs experiments remain confusing. The references cited did not explain why the constructs only abolished the repressive function of Olig2, but not the activating function. The constructs cited in the references used a subdomain of Olig2. Is it the same construct used here or is it full length Olig2? Similarly, if the inhibitory function was abolished, is the interaction with Setdb1 disrupted? Wouldn't one expect the activation form of Olig2 to induce lineage progression given its interaction with Brg1? Overall, these experiments are very confusing. While these experiments may bear many technical difficulties in nature, one possibility may be to reorganize the manuscript entirely and focus on the strength of demonstrating the function of Setdb1.

4. In all places where it labels Suv39h, it should be changed to Suv39h1 to correctly represent the gene name.

5. SuppFig 1c, "pie chart showing relative percentage of genes..." should be "showing number of genes.."

6. SuppFig 1d, "Real-time PCR analysis of Sox11 in OPC..." should be "...PCR analysis of Mag and Mbp.."

Reviewer #3:

Remarks to the Author:

The authors have carried out additional experiments and all my concerns have been fully addressed. Thus I recommend for acceptance of the manuscript.

Responds to the reviewer's comments:

Reviewer #1 (Remarks to the Author):

The authors have made a great effort to convincingly address all my questions and comments and, to my opinion, also the ones of the other reviewers. The manuscript is now a lot improved and the conclusions of the authors strengthened.

By going through their answers to the other reviewers, I however need to comment on the answer (2) to comment 1. of Reviewer #2 : There are many more HMTs than the 5 listed by the authors in their answer, but I assume that they mean 5 H3K9 HMTs. To my knowledge, there is however at least one more H3K9 HMT called PRDM2, which can also trimethylate H3K9. To fully answer comment 1. of Reviewer #2, PRDM2 expression and co-IP with Olig2 would therefore also need be checked.

Reply: Thank you for support. It's a good suggestion and we checked *Prdm2* expression in database of Barres Lab (Zhang et al., J Neurosci, 2014, PMID: 25186741). As shown below, *Prdm2* expression is downregulated when OPCs become mature. And the expression levels of *Prdm2* are comparable among astrocytes, neurons, OPCs and microglia.

Gene symbol	Astrocytes	Neuron	OPC	iOL	mOL	Microglia
Prdm2	6.3	5.8	6.8	6.2	2.5	4.8

Then we checked interaction between OLIG2 and PRDM2 in rat iOLs. As the result shown below, we barely detected interaction between OLIG2 and PRDM2. We added this data to Supplementary Fig. 2f.

Reviewer #2 (Remarks to the Author):

The revised manuscript remains largely the same as the previous version with limited attempts to provide additional evidence for the major conclusion. The additional experiments as well as the rebuttal points were insufficient and could not address the major concerns raised previously. The repressive role of Olig2 in regards to epigenetic regulation remains elusive and confusing. Several key technical details are still missing. The manuscript also bears several careless oversight. I suggest the authors to carefully proof read the manuscript.

1. One key experiment missing in the previous manuscript is the ChIP-seq of SETDB1 and its overlap with Olig2 peaks. The author claimed some technical difficulties with Setdb1 ChIP-seq and therefore proceeded with an exogenously expressed molecule through HA-tag. While exogenously expressed protein may not faithfully representing the physiological function, this experiment is a plus nevertheless. However, several key elements are missing: a) same experiments should be performed in OPC, (b) peak alignment between Setdb1 peak and Olig2 peak should be evaluated in both OPC and iOL (c) motif analysis on Setdb1 peaks in both OPC and iOL. These experiments are essential. Furthermore, if Olig2 was inducing transcriptional repression by recruiting Setdb1 upon differentiation, one would expect the enrichment of H3K9me3 around Olig2 binding site is only present in iOL but not in OPC. This is NOT what is shown in Fig 2l and 2m. In addition, to exclude the possibility that Setdb1 was recruited by other transcription factor, the co-IP experiments should be performed in the same cell type during differentiation, because these interactions are expected to cell type specific and lineage specific.

Reply: Thank the reviewer for the questions.

(1) We agree that exogenously expressed protein may not faithfully represent the physiological function. Studies from other researchers showed that overexpression of proteins which promoted myelination in OPCs would induce OPC differentiation even without T3 induction (Brg1, Yu et al., Cell, 2013, PMID: 23332759) (Chd7, He et al., Nature Neuroscience, 2016, PMID: 26928066) (Seh1, Liu et al., Neuron, 2019, PMID: 30876848). And we observed that differentiation-related genes were upregulated when SETDB1 was overexpressed, as shown blow. Therefore ectopic expression of HA-SETDB1 in OPCs won't reflect the nature of OPCs.

(2) Motif analysis on SETDB1 binding sites in iOL revealed that the consensus binding motif for OLIG2 is enriched at SETDB1 binding sites for those downregulated genes upon OPC differentiation, as the result shown below. This result is consistent with the repressive role of SETDB1 and OLIG2. We added this data to Supplementary Fig. 2k.

Motif of OLIG2 in HA-SETDB1 ChIP-seq

Down-regulated gene loci	p-value=0.01
--------------

(3) For Fig 2L and 2M, since the interaction between OLIG2 and SETDB1 is significantly augmented upon OPC differentiation as shown in Fig 2F, the enrichment of H3K9me3 around OLIG2 binding sites is also increased. However, there is still weak interaction between OLIG2 and SETDB1 even in the OPCs as shown in Fig 2F, therefore we can also observe weak enrichment of H3K9me3 in OPCs.

(4) To exclude the possibility that SETDB1 was recruited by other transcriptional factors, we performed the co-IP assays in the rat iOLs. As the result shown below, SETDB1 did not interact with these main transcriptional factors in OPC lineage cells. We added this data to Supplementary Fig. 2I.

2. One of the other major concerns is the work here presented different, in some cases, contradictory findings to the work from the Casaccia lab, who documented the role of H3K9me3 in OPC differentiation and myelination earlier (Liu et al. J Neurosci. 2015). A few examples include the effect of knocking down HMTs in OPC and the peak binding motif analysis. Given the two studies are highly relevant, the work from the Casaccia lab should be cited, and if possible, compared side by side. Discrepancies, if any, should be evaluated and discussed. The authors attributed the discrepancies between their study and Liu et al. to different species used. It is important to note that Liu et al. used both rat and mouse OPCs for their in vitro study, with ChIP-seq performed using the same species. Therefore, it is quite unlikely as both groups are examining key differentiation regulators and lineage genes. Other possibilities should also be evaluated. For example, the efficiency of HMT knockdown is ~50% by transcripts in the current study (SuppFig.2b). What about protein levels? Are

other HMTs bearing longer half-life therefore rendering them to remain functional longer than SETDB1 after silencing? When was differentiation induced after silencing? The authors claimed to focus on SETDB1 because “SETDB1 is the most dominant HMT in OPC development exactly, as shown in Fig. 2c”. This is not what is shown in Fig.2c. While the interaction between Olig2 and Setdb1 was performed under physiological conditions, the interaction between Olig2 and other HMTs were performed with exogenously expressed HMTs, and in a non-relevant cell line, 293T cells. Wouldn't one expect these interactions to be cell type-specific and lineage-specific? Finally, the level of H3K9me3 should be quantified in all Setdb1 cKO mice. For example, in SuppFig 3b, there are clearly many TdTomato+ OL that have H3K9me3. The tools and methods used for motif analysis are not discussed in the method section either.

Reply: Thank the reviewer's advice.

(1) We cited Casaccia lab's study about H3K9me3 in OPC differentiation as reviewer suggested. We speculate that the discrepancies between two studies may be attributed to the shRNA delivery methods. In Liu's paper, lentivirus was used to delivery shRNA into OPCs. However, we used electroporation assay via a Lonza Nucleofector device and OPCs were induced to differentiate 36h later. We also discussed about the discrepancies in Discussion section.

(2) We examined the knockdown efficiencies as suggested by Reviewer. As the result shown below, the knockdown efficiency is consistent with the qPCR result and all the HMTs were downregulated to the similar level. We added this data to Supplementary Fig. 2c.

(3) EHMT2 is specifically associated with H3K9me and H3K9me2 and don't catalyze H3K9me3. We examined the interactions between OLIG2 and other HMTs in iOLs. As the result shown below, OLIG2 mainly binds to SETDB1, but not other HMTs. We added this data to Supplementary Fig. 2f.

(4) We quantified the level of H3K9me3 in *Setdb1* cKO mice. As the result shown below, the level of H3K9me3 is dramatically reduced in *Setdb1* cKO mice. The remaining H3K9me3 signals in Tomato⁺ cells may be attributed to recombinant efficiency of *Setdb1*. We added this data to Supplementary Fig. 3c.

(5) We carried out Motif analysis by using HOMER. And we added this description into the method section as reviewer suggestion.

3. Using various transgenic mouse models, overexpressing constructs and demyelination model, the function of *Setdb1* in OPC differentiation and myelination is convincing, except when the authors attempt to link to the repressive function of *Olig2*. For example, as a master regulator activating many of the lineage progression genes, the activating role of *Olig2* is overwritten due to the loss of its repressive role by the lack of *Setdb1* in cKO mice. This is a very intriguing point. Why? What about the interaction between *Olig2* and chromatin remodeling enzymes as Yu et al. demonstrated previously (e.g. *Brg1*)? What happens to those loci regulated by *Olig2*'s activating role? Similarly, the VP64 constructs experiments remain confusing. The references cited did not explain why the constructs only abolished the repressive function of *Olig2*, but not the activating function. The constructs cited in the references used a subdomain of *Olig2*. Is it the same construct used here or is it full length *Olig2*? Similarly, if the inhibitory function was abolished, is the interaction with *Setdb1* disrupted? Wouldn't one expect the activation form of *Olig2* to induce lineage progression given its interaction with *Brg1*? Overall, these experiments are very confusing. While these experiments may bear many technical difficulties in nature, one possibility may be to reorganize the manuscript entirely and focus on the strength of demonstrating the function of *Setdb1*.

Reply: (1) *OLIG2*, like other transcriptional factors, could independently perform inhibition or activation functions on different loci of genome. Our study reveals that

the downregulation of genes that are repressed by OLIG2 is as essential as the upregulation of genes that are activated by OLIG2 for OPC differentiation. Loss of OLIG2's repressive function by Setdb1 deletion causes the upregulation of genes (such as SOX11) that impairs OPC differentiation, whereas the OLIG2's activation function maintain intact.

(2) We used constructs of full length OLIG2 to fuse with VP64. It's also a frequently-used strategy to construct VP64 fusion protein, such as (Olga Villamizar et al., 2020, PMID: 33532041) (Wolfgang Pita-Thomas et al., 2021, PMID: 34290335)

(3) We also explored whether VP64 could abolish the interaction between SETDB1 and OLIG2. As the result shown below, fusion with VP64 has little impact on SETDB1 and OLIG2 interaction. VP64 may work as a strong activation element to cover up the repressive role of SETDB1, instead of pushing out SETDB1.

4. In all places where it labels Suv39h, it should be changed to Suv39h1 to correctly represent the gene name.

Reply: It's our mistake. We corrected it according to the reviewer's suggestion.

5. SuppFig 1c, "pie chart showing relative percentage of genes..." should be "showing number of genes.."

Reply: It's our mistake. We corrected it according to the reviewer's suggestion.

6. SuppFig 1d, "Real-time PCR analysis of Sox11 in OPC..." should be "...PCR analysis of Mag and Mbp.."

Reply: The reviewer is right. We have corrected the sentences as the reviewer's suggestion.

Reviewer #3 (Remarks to the Author):

The authors have carried out additional experiments and all my concerns have been fully addressed. Thus I recommend for acceptance of the manuscript.

Reviewers' Comments:

Reviewer #2:

Remarks to the Author:

The revised manuscript by Zhang et al. has provided several additional evidence and technical information for their experiments, which has greatly improved the manuscript. The authors have discussed some discrepancies between their studies and previous published findings with regards to technical differences from introducing the silencing construct, electroporation vs. viral induction. This is possible as electroporation is known to be less efficient in oligodendrocyte cells, which is also suggested by the protein analysis in Suppl Fig 2c. with sub-optimal knockdown level.

However, one of the essential experiments to support the main conclusion is still missing, which is the demonstration of overlaps of Setdb1 peaks and Olig2 peaks in both OPC and iOLs. The authors have used an exogenously expressed Setdb1 to access its genomic binding loci. This is acceptable given the technical difficulties to use endogenous protein, but it needs to be demonstrated in both OPCs and iOLs. The comparison of SETDB1 peaks on Olig2 sites should be represented as Fig 2l and 2m. This is a key experiment to demonstrated the co-regulation on these genomic regions by the Olig2 and Setdb1 complex, and should perhaps be included in the main figures, not supplemental.

As an important control, could the author demonstrate that the activating role of Olig2 was undisturbed in Setdb1 cKO mice? For example, does the complex of Olig2 with chromatic modifying enzymes still form? This would be important to demonstrate the activating role and repressive role of Olig2 are independent and that they are only disrupting the repressive role of Olig2 in Setdb1 cKO mice.

It is unclear what is shown in Supp Fig 2k. Additional description would be beneficial.

Reviewers' comments:

Reviewer #2 (Remarks to the Author):

The revised manuscript by Zhang et al. has provided several additional evidence and technical information for their experiments, which has greatly improved the manuscript. The authors have discussed some discrepancies between their studies and previous published findings with regards to technical differences from introducing the silencing construct, electroporation vs. viral induction. This is possible as electroporation is known to be less efficient in oligodendrocyte cells, which is also suggested by the protein analysis in Suppl Fig 2c. with sub-optimal knockdown level.

However, one of the essential experiments to support the main conclusion is still missing, which is the demonstration of overlaps of Setdb1 peaks and Olig2 peaks in both OPC and iOLs. The authors have used an exogenously expressed Setdb1 to access its genomic binding loci. This is acceptable given the technical difficulties to use endogenous protein, but it needs to be demonstrated in both OPCs and iOLs. The comparison of SETDB1 peaks on Olig2 sites should be represented as Fig 2l and 2m. This is a key experiment to demonstrated the co-regulation on these genomic regions by the Olig2 and Setdb1 complex, and should perhaps be included in the main figures, not supplemental.

Reply: Thank the reviewer for the question. To explore the comparison of SETDB1 peaks on OLIG2 sites between OPCs and iOLs, we performed and analyzed HA-SETDB1 ChIP-seq in OPCs and iOLs. We presented the data as Fig. 2l and 2m. As the results shown below, the enrichment of SETDB1 was strengthened in loci of genes that are downregulated with enhanced OLIG2 binding upon differentiation. We added these data to the main figures.

As an important control, could the author demonstrate that the activating role of Olig2 was undisturbed in Setdb1 cKO mice? For example, does the complex of Olig2 with chromatic modifying enzymes still form? This would be important to

demonstrate the activating role and repressive role of Olig2 are independent and that they are only disrupting the repressive role of Olig2 in *Setdb1* cKO mice. Reply: It's a good suggestion. The transcriptional activating role of OLIG2 is dependent on the formation of OLIG2-BRG1 complex (Yu et al., Cell, 2013, PMID: 23332759), which was NOT disturbed in *Setdb1* cKO mice. We checked the integrity OLIG2-BRG1 transcriptional activation complex in *Setdb1* knock-down iOLs by performing Co-IP assay. As the result shown below, the interaction between OLIG2 and BRG1 was barely affected when *Setdb1* was knocked down. Master regulator could exert transcriptional activation or repression independently, recruiting different epigenetic factors to different regions of the genome. Therefore, the transcriptional activation complex of OLIG2 is intact no matter the repressive complex is disrupted with *Setdb1* deletion.

It is unclear what is shown in Supp Fig 2k. Additional description would be beneficial.

Reply: Thank the reviewer's suggestion. We added the following description into main text. "OLIG2 binding motif was identified by HOMER analysis within HA-SETDB1 peaks among loci of genes downregulated upon OPC differentiation. Letter size indicates nucleotide frequency at each position of OLIG2 binding motif."

Reviewers' Comments:

Reviewer #2:

Remarks to the Author:

The revised manuscript has addressed all my concerns. I have no further comments.

REVIEWERS' COMMENTS

Reviewer #2 (Remarks to the Author):

The revised manuscript has addressed all my concerns. I have no further comments.

Reply: Thanks for the reviewer.